



# CLASH - Climate-responsive Land Allocation model with carbon Storage and Harvests

Tommi Ekholm[1], Nadine-Cyra Freistetter[1], Aapo Rautiainen[1], Laura Thölix[1]

[1] Finnish Meteorological Institute, P.O. BOX 503 FI-00101 Helsinki, Finland

*Correspondence to*: Tommi Ekholm (tommi.ekholm@fmi.fi)

**Abstract.** The Climate-responsive Land Allocation model with carbon Storage and Harvests (CLASH) is a global land-use model that can be embedded into integrated assessment models (IAMs). It represents vegetation growth, terrestrial carbon stocks, and production from agriculture and forestry for different land-uses in a changing climate; and hence, allows the inclusion of terrestrial carbon stocks, agriculture and forestry in global climate policy analyses. All terrestrial ecosystems and
their carbon dynamics are comprehensively described at a coarse resolution. Special emphasis is placed representing the world's forests. In CLASH, vegetation growth, soil carbon stocks, agricultural yields and natural disturbance frequencies react to changing climatic conditions, emulating the dynamic global vegetation model LPJ-GUESS. Land is divided into ten biomes with six land-use classes (including forests and agricultural classes). Secondary forests are age structured. The timing of forest harvests affects forest carbon stocks; and hence, carbon storage per forest area can be increased through forest management.
In addition to secondary forests, CLASH also includes primary ecosystems, cropland, and pastures. The comprehensive inclusion of all land-use classes and their main functions allows representing the global land-use competition. In this article, we present, calibrate, and validate the model, demonstrate its use, and discuss how it can be integrated into IAMs.

## 1 Introduction

CLASH (Climate-responsive Land Allocation model with carbon Storage and Harvests)[1] is a biophysical land-use model that
describes the allocation of land to different uses, forest growth, terrestrial carbon stocks, and the production of agricultural and forestry goods globally. Global land area is divided into 10 biomes, and the area in each biome can be allocated to different uses, including agriculture, forestry, and primary ecosystems. The biophysical properties of land and vegetation are biome-specific and respond to climate change. The model has been parametrized to emulate the dynamic global vegetation model Lund-Potsdam-Jena General Ecosystem Simulator (LPJ-GUESS) (Lindeskog et al., 2021; Smith et al., 2001, 2014) in varied
and changing climatic conditions. In this article, we describe, calibrate, and validate the model and demonstrate its use.

CLASH can be embedded in economic optimization models, integrated assessment models (IAMs) in particular; or used independently to simulate the climatic impacts of land use in long-term scenarios. However, CLASH has been specifically

---

[1] Model version used in this manuscript: 2023-07-03





developed for the former purpose. In this role, CLASH represents the biophysical dynamics of land-use, while the IAM provides the rationale for *why* land should used and managed in a specific way, and also how the climate changes over time.

Three features make CLASH especially suitable for incorporation into IAMs: (1) its technical design and low computational burden, (2) global geographical coverage, and (3) comprehensive representation of land-use and carbon stocks. CLASH is computationally lightweight, linear, and has an adjustable timestep. Many IAMs are written as intertemporal optimization problems (Keppo et al., 2021), i.e., the whole modelling time horizon is considered and solved for optimum within a single optimization problem. Such models can be computationally challenging to solve, especially if the model is nonlinear.

CLASH's low computational expense facilitates its incorporation into such IAMs; and given its linear mathematical formulation, it can be incorporated also into IAMs defined as linear programming problems. However, the formulation of the optimization problem is not inherently a part of CLASH, but needs to be provided by the IAM. The underlying temporal resolution is annual, which can be flexibly adjusted to match any longer timestep, such as 5 or 10 years commonly used in IAMs.

Geographically, CLASH covers all global land area and depicts the global production of goods in agriculture and forestry. The model covers the carbon stocks of vegetation, litter and soil, and describes how they are affected by land-use and climate change. The effect of climate change on vegetation growth and soil carbon dynamics are modelled as a function of global mean temperature change and atmospheric $CO_2$ concentration, as these two variables are standard outputs of many IAMs. Together, the geographical coverage, representation of main terrestrial carbon stocks and the inclusion of climate change effects on

terrestrial ecosystems make CLASH ideal for examining land-based climate change mitigation and adaptation measures at the global scale and over long time-horizons.

Embedding CLASH in an IAM enables the optimization of land-use *and* forest management over a multi-decadal timeframe in a changing climate; and portraying the optimal contribution of the land-use sector towards the global climate change mitigation effort. This capability fills a critical, vacant niche in the model ecosystem. Towards this role, CLASH combines

features from three model types:

(1) *Dynamic Global Vegetation Models* (DGVMs), such as LPJ-GUESS (Lindeskog et al., 2021; Smith et al., 2001, 2014), can be used to depict how vegetation responds to changing climatic conditions. DVGMs are notably more detailed than CLASH. In DGVMs, however, land-use can only be depicted by exogenous scenarios. It cannot be optimized. Due to their high level of detail and heavy computational burden, DVGMs cannot be embedded in IAMs

in the way that CLASH can.

(2) *Economic partial equilibrium models of the land-use sector*, such as MAgPIE (Dietrich et al., 2019) and GLOBIOM (Havlík et al., 2018), enable the optimization of land-use within a timestep, while recursive dynamic rules portray the evolution over years. Such models represent land-use comprehensively and can be linked to IAMs (Fricko et al., 2017). However, as they stem from an agricultural economic modelling tradition, the models do not necessarily

represent comprehensively the dynamic changes in the age structure of forests (GLOBIOM); or allow for intertemporal optimization (MAgPIE and GLOBIOM). Both of these features are needed to enable the dynamic optimization of forest management when, e.g., studying global resource use or climate policy questions.

(3) *Forest sector models*, such as the GTM (Sohngen et al., 1999), include forest age structure and rely on intertemporal optimization as the solution concept. Although forest sector models have been linked to IAMs (Favero & Mendelsohn,

2014; Sohngen & Mendelsohn, 2003; Tavoni et al., 2007) and explicit, age-structured representations of forests have





been built into IAMs (Siljander & Ekholm, 2018); they do not depict land-use comprehensively as they (by definition) focus on the forest sector.

By combining features from the above three model types, CLASH provides a comprehensive, climate-responsive depiction of global land-use. This comes at the expense of detail. Ecosystem characteristics, land-use competition, agriculture and forest
management are described in less detail than in models focusing on each aspect individually.

This paper provides a 'proof of concept' description for the model and how it could be utilized. The current parametrizations of CLASH are based on non-bias-corrected climate data, however, which can lead to some deviation from reality regarding vegetation characteristics. New parametrizations based on bias-corrected data will be provided with subsequent model versions, and should be used for analyses.
The rest of the article is organized as follows. In section 2, we describe the structure of the model and the modelling of vegetation, soils, and crop and timber yields. In section 3, we present the calibration of the model and in section 4, the validation of this calibration against the global terrestrial carbon stocks projected by LPJ-GUESS. As a demonstration of the model, we analyze in section 5 how different demand scenarios for agricultural and forestry products affect the possibilities for enhancing terrestrial carbon stocks. Last, in section 6, we further discuss the integration of CLASH with IAMs and the possible uses the
model might have.

## 2      Model structure

### 2.1     Dimensions and variables

The basic dimensions of CLASH are biomes $b \in B$ into which global land area is divided, land-use categories $u \in U$, and timesteps $t \in T$. The basic timestep resolution is annual, but most use cases – especially when combined with an IAM – require
using a multi-year timestep, such as 10 years. The age structure of secondary forests is modelled though age classes $a \in A$.

The variables describe:

- land area (by biome, land-use type and timestep)
- carbon stocks in vegetation, (by biome, land-use type and timestep)
- carbon stocks in woody and herbaceous litter and soil (by biome and timestep)
- areas of forest clearing (by biome, age-class and timestep)
- harvested crops and wood (by biome and timestep)
- headcount and product yield of agricultural animals (by timestep)
- $CH_4$ and $N_2O$ emissions from agriculture (by biome and timestep)

### 2.2     Ecological and land-use modules

CLASH consists of a land-use module and an ecological module. The land-use module contains the variables and equations for land allocation, terrestrial carbon stocks and harvesting; and is the part that can be integrated into an IAM. The ecological





module is used to calibrate the land-use module's parameters based on the trajectories of climate change. The ecological module is not designed to be integrated into the IAM, as it would violate the linear formulation required by many IAMs.

Climate change affects vegetation growth; and this impact differs across biomes through local climatic factors, such as temperature and precipitation. In CLASH, growth is determined by ecological parameters that are modelled as a function of global mean temperature and carbon concentration, as these variables standard outputs of many IAMs. Hence, changes in these variables serve as a proxy for the changes on local climatic factors. Notably, the biomes are large and likely to cover heterogeneous response to climate change (e.g., some parts of a tropical biome may become drier, while others get wetter). Therefore, the climate-induced changes on growth depict changes in the average conditions within each biome.

The division into two separate modules was done to satisfy two conflicting model design requirements: (1) the land-use module must be linear and computationally lightweight, and (2) ecological conditions must respond to climate change. Some ecological parameters depend nonlinearly on climatic conditions, but including this climate-dependence in the land-use module would make the model nonlinear and involve more complex calculations. Instead, linearity is maintained in the land use module through fixed, time-varying parameter values to depict vegetation growth, disturbances, yields and carbon dynamics, which change over time according to a predetermined climate change scenario. When CLASH is integrated into and IAM, one can iteratively run the IAM and then re-calibrate the parameters with the ecological module, using the climate trajectory in the IAM's solution. The procedure is repeated until consecutive solutions converge, and the ecological parameter values align with the climate trajectory. Whether the iterative procedure is necessary depends on the IAM and scenario design.

### 2.3 Land-area allocation

Global land area is divided into 10 biomes based on the USDA Major Biomes classification (Reich & Eswaran, 2020). Biomes of marginal importance to agriculture, forestry and carbon stocks – such as 'ice' and 'permafrost' – are aggregated into a single 'unproductive' class. We keep the geographical boundaries of biomes constant over time, even as the climate changes. Instead, the ecological parameters depicting vegetation growth, disturbances, agricultural yields and carbon dynamics respond to climate change. A map of the applied biome classification is presented in Figure 1.

Two requirements guided the choice of classification: *conciseness* (i.e., having only a relatively small number of biomes) and *relative homogeneity* (i.e., keeping the variation in growth and carbon dynamics within each biome small). These are conflicting requirements, as greater conciseness leads to less homogenous biomes. The USDA major biomes classification was chosen as the basis for the biomes in CLASH, as it divides the world to relatively few biomes, but which were more homogenous than with alternative classifications, such as the Köppen-Geiger climate classification or Holdridge life-zones.

The land-use classes in CLASH are based on the Land-use Harmonization dataset (LUH2) (Hurtt et al., 2020). The classes are cropland, pasture (including rangeland), primary ecosystems (including primary forest and primary non-forest), secondary forest, secondary non-forest and urban land. Primary ecosystems are ecosystems that have not been notably altered by direct human disturbance. Secondary forests have been logged at least once or have been established on previously unforested land.

Secondary non-forest is land that is not actively used but has been subject to human land-use. Land cannot be converted back

into primary ecosystems. Hence, once cleared, primary ecosystems cannot be regained.

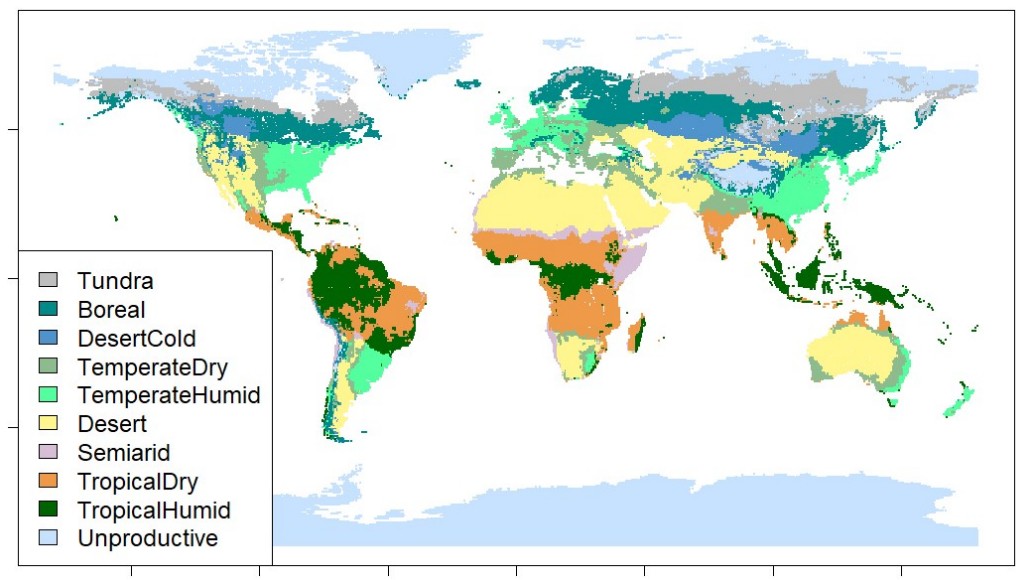

**Figure 1. Biome classification used in the CLASH model.**


The land-use module allocates land area to different uses, which affects the production quantities of goods (food and feed crops, wood, etc.) and carbon storage in vegetation and soil. Land-use in each biome, $b$, is constrained by the biome's total land area $A_b$ (million km$^2$). This area is distributed between land-uses, $u$, and the area of land-use $u$ in period $t$ is $A_{b,u,t}$. Hence, for all biomes $b$ and timesteps $t$,

$$\sum_u A_{b,u,t} = A_b. \tag{1}$$

Secondary forests ($u = secdf$) are further divided into age-classes. The width of each age-class is the same as the model time step (e.g. 10 years). All secondary forest area, $A_{b,secdf,t}$, must belong to one age-class, $\hat{A}_{b,secdf,t,a}$. Hence,

$$\sum_a \hat{A}_{b,secdf,t,a} = A_{b,secdf,t}. \tag{2}$$



Between time periods, secondary forest can age, be harvested, or get destroyed by disturbance events like forest fires. Aging forest land area will shift to the next age-class in the following time period. Area cleared by harvests or destroyed by forest disturbances is allocated to the youngest age-class in the next period if replanted, or converted into other land-use and thereby

subtracted from the secondary forest area. Other land converted to secondary forest is added to the youngest age-class.

### 2.4 Vegetation carbon stocks

Vegetation carbon stocks (GtC) are calculated by multiplying land area $A_{b,u,t}$ (million km$^2$) by vegetation carbon density $d_{b,u,t}$ (kgC m$^{-2}$). Vegetation carbon densities for all land-uses across biomes are projected by the ecological module based on global mean temperature and atmospheric $CO_2$ concentration scenarios provided to the module.

Vegetation on *cropland* and *pastures* is short-lived compared to the model time step, and hence assumed to regenerate within each model period. Cropland vegetation is represented by an aggregate crop that reflects the weighted-average properties of all crops cultivated in the biome. Likewise, pasture vegetation is depicted by representative grasses. As the vegetation regenerates frequently, the amount of vegetation and its carbon density in period $t$ is solely determined by current growth conditions (i.e., not on pre-existing vegetation stock or past growth conditions). The growth conditions in biome $b$ depend on

the global mean temperature, $T_t$, and the atmospheric $CO_2$ concentration, $C_t$.[2] The dependence is characterized by the function:

$$d_{b,u,t} = \alpha_{b,u} + \beta_{b,u} \, T_t + \gamma_{b,u} \, C_t, \tag{3}$$

where $\alpha_{b,u}$, $\beta_{b,u}$, and $\gamma_{b,u}$ are parameters estimated by fitting the function to data from LPJ-GUESS simulations.

Similarly, the growth of *secondary forests* reflects the average properties of forests in each biome. Unlike vegetation on cropland and pasture, trees are long-lived and the carbon density depends on the stand age and the climatic conditions the trees have grown in. Relatedly, forest growth depends on the growth conditions characterized by the current climate and the stand's

current state (characterized by the growth history and thus the past climate).

For a stand currently in age-class $a$, the next-period carbon density $d_{b,secdf,t+1,a+1}$ depends on its current density $d_{b,secdf,t,a}$ and the relative growth rate, $g_{b,secdf,t,a}$:

$$d_{b,secdf,t+1,a+1} = d_{b,secdf,t,a}(1 + g_{b,secdf,t,a}) \tag{4}$$

where

$$g_{b,secdf,t,a} = d_{b,secdf,t,a}^{\,\delta_b + \varepsilon_b d_{b,secdf,t,a}} (\eta_b + \theta_b \, T_t + \kappa_b \, C_t + (\lambda_b + \mu_b \, C_t)T_t^2). \tag{5}$$

---

[2] The effect of global warming on average temperature and precipitation is not uniform across biomes. However, changes in local conditions are driven by the increase in the global temperature anomaly. The increasing atmospheric $CO_2$ may enhance growth through $CO_2$ fertilization.





Parameters $\delta_b$, $\varepsilon_b$, $\eta_b$, $\theta_b$, $\kappa_b$, $\lambda_b$, and $\mu_b$ are estimated from LPJ-GUESS simulations of forest growth in various scenarios of

changing climate (see Section 3).

*Primary ecosystems* encompass primary forest and primary non-forest. Their vegetation is long-lived. However, unlike in the

case of secondary forests, the age structure (and, hence, the growth and disturbance dynamics) of primary ecosystems need

not be modelled explicitly.[3] The carbon density of primary ecosystems, $d_{b,primary,t}$, is a landscape-level average that accounts

implicitly for the age structure and the growth conditions of primary ecosystems. Its dependence on climatic conditions is

characterized by the function of the same form as in equation (3).[4]

*Secondary non-forest* is a small but diverse category of land. It contains areas that are recovering from human influence

including, for example, deforested land that is not claimed for another use and abandoned croplands and pastures. Hence, the

carbon density of secondary non-forest varies notably depending on local climatic conditions, earlier land-use, and the degree

and time since the human influence on them. These factors make the modeling of the associated vegetation and carbon stock

very difficult. As the composition of vegetation on secondary non-forest land is not specified in the LUH2 dataset, we assume

due to lack-of-data that vegetation is growing naturally in these areas. This possibly overestimates the amount of biomass and

stored carbon to some degree.

*Urban areas* cover currently only 0.4% of global land area (Hurtt et al., 2020). As vegetation carbon stocks in urban areas are

insignificant compared to the global total, they are omitted from CLASH. As urban areas do not contribute to carbon storage

or producing agricultural and forestry products in CLASH, the urban area needs to be fixed to an exogenous scenario in a

typical use-case of the model.

## 2.5    Litter and soil carbon stocks

Litter and soil carbon stocks (measured in GtC) are modelled through dynamic stock equations that account for their

accumulation, decay into atmosphere, and transfer of carbon from the litter to the soil stock. Each biome *b* has separate litter

and soil carbon stocks for woody and herbaceous matter, $L_{b,k,t}$ and $S_{b,k,t}$, distinguished by the subindex $k \in \{woody, herb\}$.

This distinction allows accounting for differences in their decay. The woody matter accumulates from primary ecosystems,

---

[3] The age structure of secondary forests is determined by harvesting patterns (which depend on human behavior and may therefore differ between model runs) and natural disturbances (which occur at exogenously given rates). Enabling the optimization of harvests requires explicitly modelling age structure. The age-structure of primary ecosystems, on the contrary, is not affected by harvests. If the land is cleared by humans, the ecosystem is no longer considered *primary*. Hence, the age structure of primary ecosystems solely by disturbances and natural mortality.

[4] This formulation does not (fully accurately) account for the growth and disturbance history of primary ecosystems (which is linked to the historical development of the climate). However, as growth conditions and disturbance regimes change fairly gradually, the error caused by adopting this (notably simpler) formulation for primary ecosystems (than secondary forests) is small.





secondary forests and secondary non-forests; and herbaceous matter from croplands and pastures. As these stocks are not directly linked to the land area under each land-use category, land-use change does not affect existing stocks, but only the accumulation of carbon.[5]

Vegetation generates litter. Its amount is defined as the difference between the annual net primary production (NPP) and the annual change in the vegetation carbon density. Additionally, in forests, harvests produce logging residues that increase the influx of woody litter; and on cropland, harvests remove a part of the carbon fixed by NPP, reducing the litter carbon influx. The total annual litter carbon influx is denoted by $I_{bkt}$.

The fraction $\left(v_{b,k} + \xi_{b,k}\, T_t\right)$ of the litter stock decays into the atmosphere annually. Here, $v_{b,k}$ is the base decay rate for biome

$b$ and $\xi_{b,k}\, T_t$ represents the effect of climate change on the litter decay. A fraction $\rho_{b,k}$ of litter carbon is transferred to the soil carbon stock. Carbon that is not released into the atmosphere or transferred into soil remains in litter. Analogously, the fraction $\sigma_{b,k} + \tau_{b,k}\, T_t$ of soil carbon decays annually to the atmosphere. This leads to the dynamic equations for litter and soil carbon stocks:

$$L_{b,k,t+1} = L_{b,k,t}\left(1 - v_{b,k} - \xi_{b,k}\, T_t - \rho_{b,k}\right) + I_{b,k,t}, \qquad (6)$$

$$S_{b,k,t+1} = S_{b,k,t}\left(1 - \sigma_{b,k} - \tau_{b,k}\, T_t\right) + \rho_{b,k}\, L_{b,k,t}.$$

### 2.6    Forest disturbances

Forest fires are the only natural disturbance in CLASH at the moment. Fires are modelled as stand-replacing disturbances. A certain share of secondary forest area in each age class burns every year, and this average fire probability changes over time with the climate. Fires also affect primary ecosystems, but the effect is not explicitly modelled: the disturbance regime and climate-induced changes are implicitly accounted for in the carbon density of primary ecosystems.

The fire probability was modelled to depend on the global mean temperature (which drives changes in local temperature and

precipitation) and the $CO_2$ concentration ($CO_2$ fertilization affects forest growth, which affects fire probability through fuel load). The linear relationship between fire probability and the climate variables is equivalent to equation (3), and the parameters of this equation were estimated from LPJ-GUESS simulations for natural forests.

---

[5] Should the litter and soil carbon stocks explicitly represent the stocks in each land-use class $u$, any change in the area of a land-use class should be also reflected in these stocks. Then, a change from forest to pasture, for example, would require transferring litter carbon from forests to pastures, but this would affect the decay of this stock, (see e.g. Rautiainen et al., 2017). Alternatively, one could account for the land-use change history, but this would complicate the model. Due to these considerations, the litter and soil carbon stocks were chosen to not to explicitly represent the carbon stored in each land-use class. After choosing this relative independence from the land-use classes, two stock types (woody and herbaceous) already provided sufficient accuracy for this model's scope.





### 2.7     Food crop and wood harvests

Food crops are harvested from cropland. Each biome's average crop yield $y_{b,t}$ (kg$_{DM}$ m$^{-2}$ year$^{-1}$) is a weighted average of the
yields of the five plant functional types (PFTs) grown on cropland (C3 annuals, C4 annuals, C3 perennials, C4 perennials, and
C3 nitrogen fixing plants), and accounting for the share of irrigated and rainfed crops at each location. The average crop yield
is represented by the same functional form as was used with vegetation carbon density, in equation (3). Total crop harvest in
each biome is the product of average crop yield and cropland area.

Wood is harvested from secondary forests and cleared primary ecosystems by clear-cutting. Wood harvests depend on
harvested area and the stocking density (i.e. stem volume) $v_{b,u,t,a}$ (m$^3$ ha$^{-1}$) of the harvested forests; which varies across biomes,
and also across age classes for secondary forest. The stem volume is calculated by multiplying the carbon density, $d_{b,u,t,a}$ (kg
m$^{-2}$) with the conversion factor, $\gamma_b$ ((m$^3$ ha$^{-1}$)/(kg m$^{-2}$)).[6] Hence,

$$v_{b,u,t,a} = \gamma_b d_{b,u,t,a}. \tag{7}$$

The conversion factors applied in CLASH are based on data from FAO's Global Forest Resources Assessment 2020 country
reports (FAO, 2023). Their values are displayed in Table 1.[7]
Wood harvesting generates three timber grades: logs, pulpwood and energy wood (m$^3$ year$^{-1}$) and forest residues (kg$_{DM}$ year$^{-1}$),
which includes all biomass not covered by the aforementioned categories. The largest parts of large stems qualify as logs and
may be used for timber. Pulpwood includes small stems, thin parts of large stems, and large branches. Energy wood are
treetops, very small stems and small branches. Residues may be harvested or left on site, in which case the carbon in them
enters the woody litter carbon pool.


---

[6] As the carbon content of biomass is roughly 50%, dividing $d_{b,u,t,a}$ by 0.5 gives the amount of biomass in the forest. Biomass
(kg/m$^2$) is converted into stem volume (m$^3$/ha) by multiplying by a biome-specific ratio of stem volume to biomass. These two
operations can be combined into a single conversion factor, $\gamma_b$ ((m$^3$/ha)/(kg/m$^2$)), that directly converts carbon density into
stocking density.

[7] The conversion factor for each biome is based on data from a representative country that is predominantly located within on
biome. The geometry of trees within these countries is assumed to roughly represent geometry of trees within the biome. (Data
for other countries was checked to verify the correct magnitude of the conversion factors). The conversion factors are based
on estimates of average growing stock (m$^3$ ha$^{-1}$ over bark, reported in Section 2A of each report) and forest biomass (t ha$^{-1}$,
reported in Section c of each report). The conversion factors also incorporate a unit conversion from kg m$^{-2}$ of biomass to m$^3$
ha$^{-1}$ of stem volume.





**Table 1. Conversion factors for translating vegetation carbon density (kg/m$^2$) into stem volume (m3/ha)**

| biome | conversion factor $(m^3\ ha^{-1})/(kg\ m^{-2})$ | assumptions | wood carbon content $(tC\ m^{-3})$ | approximate residue fraction (unitless) |
|---|---|---|---|---|
| Boreal | 28.4 | value from Finland (2020) | 0.190 | 0.46 |
| Tundra | 28.4 | adopted from boreal biome | 0.190 | 0.46 |
| Desert Cold | 27.0 | value for Mongolia (2020) | 0.190 | 0.49 |
| Temperate Dry | 21.4 | value for Greece | 0.215 | 0.54 |
| Temperate Humid | 29.5 | value from Japan (2017) | 0.215 | 0.36 |
| Desert | 5.6 | value from Namibia (2020) | 0.215 | 0.88 |
| Tropical Dry | 21.1 | value from Tanzania (2020) | 0.240 | 0.49 |
| Tropical Humid | 22.8 | value from Brazil (2020) | 0.240 | 0.45 |
| Semiarid | 17.0 | value from Kenya (2020) | 0.215 | 0.63 |
| Unproductive | 28.4 | adopted from boreal biome | 0.190 | 0.46 |

The division of stem volume into timber grades depends on the stem volume $v_{buta}$. Stands with very small trees and low

stocking density provide only energy wood; stands with large trees and high stocking density provide mostly logs and some

pulpwood.[8] Let $\sigma_i(v_{buta})$, where $i \in \{energy, pulp, logs\}$, denote the share of each timber grade as a function of stocking

density. We assume the following breakdown:

$$\sigma_{energy} = \begin{cases} 1 & when\ v_{b,u,t,a} < 20, \\ 1 - 0.0085\,(v_{b,u,t,a} - 20) & when\ 20 \leq v_{b,u,t,a} \leq 120, \\ 0.15, & when\ 120 < v_{b,u,t,a}, \end{cases} \qquad (8)$$

and

[8] The functions described here are based on expert judgment and roughly characterize the development of the assortment shares
in boreal forests. We assume that the connection between stocking density and assortment shares is roughly similar across
biomes. Hence – lacking biome-specific data for the calibration – we apply the same functions to all biomes. The accuracy of
future model versions maybe improved by developing biome-specific functions.





$$\sigma_{logs} = \begin{cases} 0 & when\ v_{b,u,t,a} < 80, \\ (1 - \sigma_{energy})0.00425\ v_{b,u,t,a} & when\ 80 \leq v_{b,u,t,a} \leq 280, \\ (1 - \sigma_{energy})0.85, & when\ 280 < v_{b,u,t,a}, \end{cases} \tag{9}$$

and

$$\sigma_{pulp} = 1 - \sigma_{energy} - \sigma_{logs}. \tag{10}$$

Let $\vartheta_b$ and $\rho_b$ denote respectively the carbon density of wood in biome $b$ (Table 1)[9] and the fraction of total biomass carbon contained in residues. As residues contain all carbon not contained in the stems, we define

$$\rho_b := \frac{d_{b,u,t,a} - 10^{-1}\vartheta_b v_{b,u,t,a}}{d_{b,u,t,a}} = 1 - \frac{\vartheta_b \gamma_b}{10}. \tag{11}$$

### 2.8   Livestock

CLASH includes four kinds of livestock, that only produce meat: beef cattle, pigs, broiler chicken and "shoats" (aggregated sheep and goats); as well as dairy cattle that produces milk *and* beef, and laying hens that produce eggs. The livestock variables
covered in the model are headcount per animal type (millions), production of each animal product (Mt year$^{-1}$), and $CH_4$ and $N_2O$ emissions from enteric fermentation and manure management (Mt year$^{-1}$).

Table 2 displays the animal products' modelling assumptions for global current herd size and per-head pasture and feed requirements (for modelling the land use for animal husbandry), product yields (for modelling the supply of animal products), GHG emissions factors (for modelling the animals' climatic impacts) and average-animal characteristics (provided for
reference).

---

[9] The carbon density of a cubic meter of wood is the product of (1) its mass, and (2) the carbon content of wood. Different wood species have a different dry weight. Hence, the average dry mass of a cubic meter of wood varies between biomes, depending on species composition. Biome-specific values of average wood mass are not readily available. Hence, we use estimates from Rautiainen (unpublished) for boreal, temperate and tropical and apply the most appropriate estimate to each biome. The carbon content of wood is approximately 0.5 tC/t$_{DM}$.





**Table 2. Data and Assumptions used in the livestock module.**

|  | Herd size in 2020 | Animal weight [a] | Pasture use [b] | Yearly feed use | GHG emission factors [d] | | Life-time | Product yield | Production |
|---|---|---|---|---|---|---|---|---|---|
|  | million heads | kg | m² per head | kg DM per head | kg CH₄ per head | kg N₂O per head | years | kg per head per year | Mt per year |
| **Beef** | 953 | 435 | 15982 | 124.2 | 58.9 | 0.943 | 2.8 | 56.4 | 53.8 |
| **Beef (dairy cows)[c]** | 573 | *435* | 605.6 | *137.3* | *96.2* | *1.100* | *6.0* | 26.4 | 15.1 |
| **Shoat** | 2390 | 37.3 | 3833 | 1.3 | 7.12 | 0.251 | 1.4 | 10.9 | 26.1 |
| **Pork** | 953 | 155 | 10.6 | 309.7 | 2.86 | 0.283 | 0.5 | 140.0 | 133.4 |
| **Chicken** | 33100 | 3.0 | 0.6 | 14.6 | 0.009 | 0.009 | 0.5 | 3.7 | 123.1 |
| **Whole Milk[e]** | 573 | 444 | 10861 | 137.3 | 96.2 | 1.100 | 6.0 | 1548 | 886.9 |
| **Eggs[e]** | 7900 | 3.0 | 2.8 | 13.9 | 0.135 | 0.009 | 2.0 | 11.2 | 88.6 |

[a] *Liveweight*

[b] *Total global in-use pasture area assumed to be 2.1 billion hectares (Mottet et al., 2017; Poore & Nemecek, 2018).*

[c] *Beef from culled dairy cows, shared values across the two sub-systems are repeated and printed in italic*

[d] *Emissions from manure management and, for ruminants, enteric fermentation*

[e] *Demand and product yield refer to the product (milk or eggs) and the rest to the product delivering animal (dairy cow or laying hen)*

The annual product yields of cow milk and chicken eggs per head (i.e., per producing animal) were obtained from (FAO, 2023). Similar (albeit theoretical), annual product yields of meat per head were calculated for meat producing animals as follows. First, the boneless meat yield per animal was estimated by multiplying its average liveweight (cattle: Dong et al., 2006; others: Gavrilova et al., 2019) by the share of boneless meat of the animal's mass (Knight & Rentfrow, 2020; Wilfong & O'Quinn, 2018). Second, the average slaughter age was approximated based on the size of the global herd and the number

of animals slaughtered annually (FAO, 2023). Finally, the annual product yield of meat per head was calculated by dividing the boneless meat yield by the average slaughter age.

The livestock carbon stock is insignificant, around 0.1 Gt C (Bar-On et al., 2018), and is therefore omitted from the model. The emission factors for methane and nitrous oxide were obtained respectively from (Gavrilova et al., 2019) and (Jun et al., 2000). Pasture use was obtained from (Mottet et al., 2017; Poore & Nemecek, 2018). A biome-specific pasture use per animal

was calculated from this global average by using the NPP of pastures in each biome, so that the biome-specific pasture use weighted with the pasture area in year 2020 matches the global average pasture use per animal, as presented in Table 2. Yearly feed use per head was obtained from (Mottet et al., 2017).[10] The crops consumed as animal feed are produced on cropland, which implies that livestock also requires cropland to produce the necessary feed.

---

[10] Supplementary material, Table SI 2 in Mottet et al. (2017).



## 3    Data and model fitting

### 3.1    LPJ-GUESS

To estimate the parameters of CLASH that define growth, disturbances, yields and carbon dynamics in each biome, we used data generated by the LPJ-GUESS model (Smith et al. 2001, 2014; Lindeskog et al. 2021) run globally with a 2°×2° grid in different climatic scenarios. LPJ-GUESS is a second-generation dynamic global vegetation model (DGVM) which has been optimised for regional to global applications. It includes a detailed representation of forest ecosystem composition and stand dynamics. It can simulate, for example, vegetation growth and succession (Smith et al., 2014) and vegetation shifts under future climate scenarios (Hickler et al., 2012). A detailed description of LPJ-GUESS is available in Smith et al. (2001). We used LPJ-GUESS version 4.0 with global PFT's.

The model simulates potential vegetation as a mixture of 19 plant functional types (PFTs) which compete with each other for light, space and soil resources in each simulated grid cell. Each PFT is characterized by growth form, phenology, photosynthetic pathway ($C_3$ or $C_4$), bioclimatic limits for establishment and survival. Additionally, woody PFTs are characterized by allometry. In "cohort mode", all individuals of a given age cohort are assumed identical (Knorr et al., 2016). The ecosystem processes are updated daily but carbon allocation is only updated annually. Crop sowing and harvesting dates are determined dynamically based on local climatology (Lindeskog et al., 2013).

Biomass-destroying disturbances are turned off, but wildfire probability is modelled prognostically based on weather, fuel continuity (litter), and human population density using SIMFIRE-BLAZE model where SIMple fire model (SIMFIRE) calculates total burned area (Knorr et al., 2014) with total fire carbon-flux calculated from BLAZE (BLAZe-induced land-biosphere-atmosphere flux Estimator) (Rabin et al., 2017).

### 3.2    Case setup: runs and climate scenarios

We ran 48 global simulations with LPJ-GUESS, varying $CO_2$ concentration and climate scenarios from different climate models. The variations are presented in Table 3. The purpose of running different climate and $CO_2$ scenarios independently of each other was to distinguish between the effects climate change and $CO_2$ fertilization. Climate scenarios from three climate models were used to assess the results' sensitivity to model choice. LPJ-GUESS simulations began with a 500-year spin-up, and after that, the actual simulations were run from 1900 to 2100. Model defaults for irrigation, fertilization and other cropland management options are used (Lindeskog et al., 2013; Olin et al., 2015).

The climatological data driving LPJ-GUESS is from the Coupled Model Intercomparison Project sixth phase (CMIP6) simulations (Eyring et al., 2016) in the Earth System Grid Federation database. We used temperature, precipitation and solar radiation from three Earth System Models (ESMs): EC-Earth3, CanESM and MPI-ESM. Citations of specific model variants and datasets are provided in Table A1 in the Appendix. These three model variants were chosen, as they give rather different results in terms of global mean temperature and precipitation: CanESM produces higher temperature and precipitation, MPI lower temperature and precipitation, and EC-Earth is between these. The datasets have been interpolated to 2°×2° grid by





Climate Data Operators (CDO) using bilinear interpolation. Climate datasets used with LPJ-GUESS to parameterize the current version of CLASH were not bias-corrected. Biases in ESM results can have a large influence on ecosystem and carbon cycle modelling (Ahlström et al., 2017), but correcting for them can also introduce new uncertainties to scenarios of future climate (Maraun et al., 2017). Although averaging over large geographical areas is likely to reduce the biases' effect on CLASH

parametrization, potential model users are advised to use parametrizations based on bias-corrected data, which we provide with subsequent model versions.

Each LPJ-GUESS simulation was based on one of two alternative climates: a warmer future climate (scenario SSP2-4.5) or colder historical climate (climate from years 1901-1930, randomly sampled). Likewise, two $CO_2$ concentration pathways were used: the SSP2-4.5 scenario or a constant concentration of 310 ppm. These four variations enable separating the effects of

climate change and $CO_2$ fertilization when parametrizing the ecological module in CLASH. The three ESMs, on the other hand, provide three distinct parametrizations for CLASH, as averaging results from disparate models did not seem meaningful. In the following, we focus on the EC-Earth parametrization for brevity. The main climate variables of temperature and precipitation in each biome are presented in Appendix A.

Forests, crops and pastures were simulated separately at a global grid resolution of 2°×2°. That is, in forest simulations, only

forest was grown at all the grid points. In crop simulations different crop types, and in pasture simulations only grass was grown. To parametrize forest growth as a function of stand age, the forest simulations included forest stands planted in 20-year intervals from 1900 to 2000. In addition, one set of simulations with LUH2 land-use (Hurtt et al., 2020) was run for primary ecosystem parametrization and model validation purposes.

**Table 3. Scenario specifications for the LPJ-GUESS simulations.**

|  | Variations |
|---|---|
| **Earth System Model** | EC-Earth3, CanESM, MPI |
| **Climate scenario** | SSP2-4.5, Historical 1901-1930 |
| **CO2 scenario** | SSP2-4.5, Constant 310 ppm |
| **Modelled land-use** | Forest, Crops, Pasture, LUH2 |

### 3.3    Parameter fitting procedure

To parametrize the functions presented in section 2, we used LPJ-GUESS output variables (such as vegetation carbon densities, litter and soil carbon stocks, NPP, crop yields, and annual forest fire probabilities) as dependent variables and climatological drivers (global mean temperature and $CO_2$ concentration) from the related climate scenarios as independent variables. All data were on an annual level. Three separate CLASH parameterizations were made, corresponding to the LPJ-GUESS runs driven

with the EC-Earth3, CanESM and MPI climate scenarios. The separate CLASH parametrizations can be used to represent some of the variation that exists between ESMs regarding future climate change, and how these variations affect vegetation growth and terrestrial carbon stocks.



For static, linear equations (e.g. equation (3)), ordinary least-squares fitting was used. For dynamic equations (equations (4) and (6)), the parameters were fitted by minimizing sum of the squared errors between the LPJ-GUESS result and values simulated using the fit over the whole timeframe (1900-2100). As this minimization problem is possibly non-convex, the parameter fitting was done in two steps: first using a global optimization algorithm to find a relatively good parametrization, then using this as a starting point for a local optimization algorithm to find the exact optimum.

To find a suitable form for each function, we started with simple (e.g., linear) representation. If this was not sufficient to replicate the LPJ-GUESS results with sufficient accuracy, additional terms were added to the function. In the case of more complex formulae, particularly the relative forest growth of equation (5), a number of functional forms were tested in each case to ensure a suitable fit. The variations included e.g. polynomial, exponential and power representation for the temperature effect; or the inclusion or exclusion of interaction-effects between the temperature and $CO_2$ concentration. The objective of the fitting procedure was to find functions that roughly emulate LPJ-GUESS. The final formulations are what was presented in section 2.

### 3.4    Fits of forest growth and fires

The development of forest carbon density with stand age in a changing climate is shown in Figure 2. The carbon density modelled with LPJ-GUESS is compared to the carbon density simulated using equation (4) with the fitted parameter values. The parametrization emulates the original LPJ-GUESS results well in all biomes and climate scenarios.



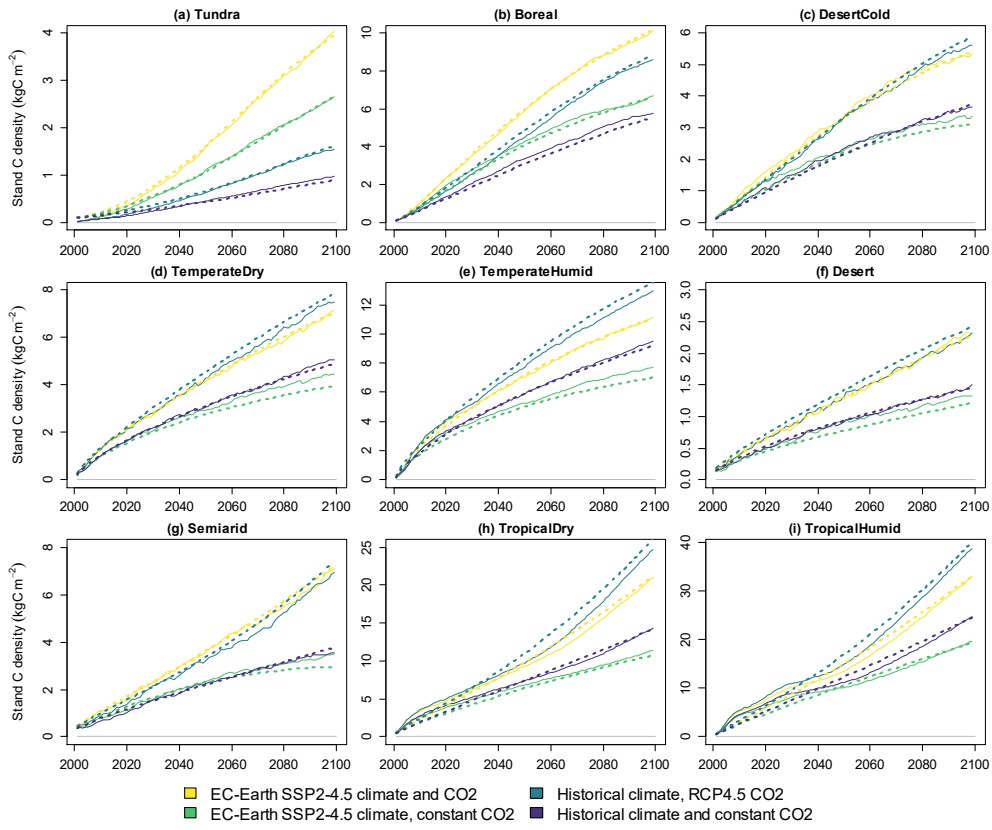


**Figure 2. Forest stand carbon density in the four simulated climate scenarios with a stand planted in 2000. Solid lines indicate LPJ-GUESS simulations and dashed lines indicate functions fitted to the results. The proximity of solid and dashed lines of the same color indicates the goodness of fit. If the lines are close, CLASH emulates LPJ-GUESS well.**

Forest growth, and how climate change affects it, varies across biomes. In colder biomes of Tundra and Boreal a warming climate increases forests growth considerably; whereas the opposite is true, although to a lesser degree, for the warmer biomes, particularly for Temperate Humid, Tropical Dry, and Tropical Humid. Higher atmospheric $CO_2$ concentrations improve growth considerably in all biomes due to the $CO_2$ fertilization effect (Walker et al., 2021). This effect is relatively stronger, in the warmer biomes than in the colder ones; which conforms with earlier analyses with LPJ-GUESS (Hickler et al., 2008).

Annual probabilities of forest fire from LPJ-GUESS and the fitted parametrization for an equation of the form (3) are displayed in Figure 3. The fitted functions capture the overall level and in most cases the trend in the incidence of forest fires. Fire probabilities depend strongly on the biomes' climate. Generally, dry and warm biomes have more recurring fires than cold and wet ones. Tundra experiences a major increase in fire probability due to a warming climate. However, in some biomes, climate





change doesn't affect fire prevalence strongly, and the changes are more driven by land-use change (Knorr et al., 2016). This

effect is not captured by the explanatory variables of equation (3).

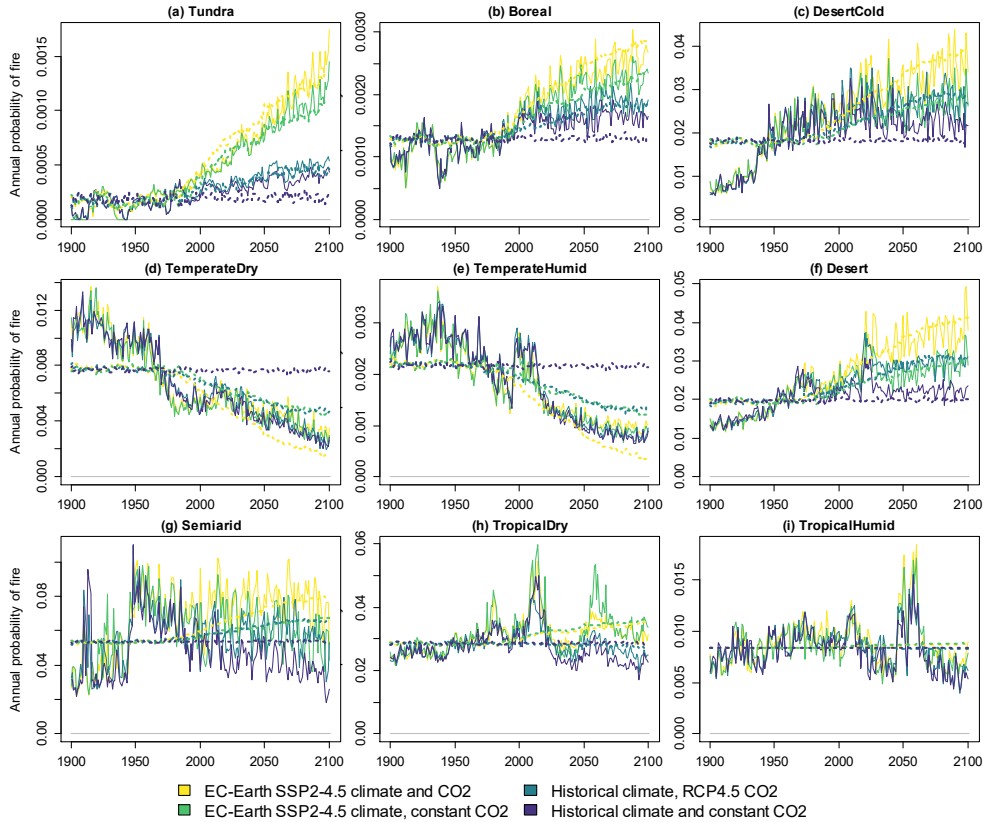

**Figure 3. Average forest fire return times in the four simulated scenarios. Solid lines indicate LPJ-GUESS simulations and dashed lines indicate functions fitted to the results.**

### 3.5    Fits of vegetation carbon stocks

The carbon densities of natural vegetation, cropland and pastures are presented respectively in Figure 4, Figure 5 and Figure 6. The linear model of equation (3), with temperature and $CO_2$ concentration as the explanatory factors, performs well in depicting the overall trends of the three vegetation types across scenarios and biomes. Cropland and pasture vegetation exhibit notable variation between consecutive years, which is not captured well by the statistical fit. However, as CLASH is primarily intended to be used at a 5 or 10-year timestep, the inability to model annual fluctuations is not a major concern.



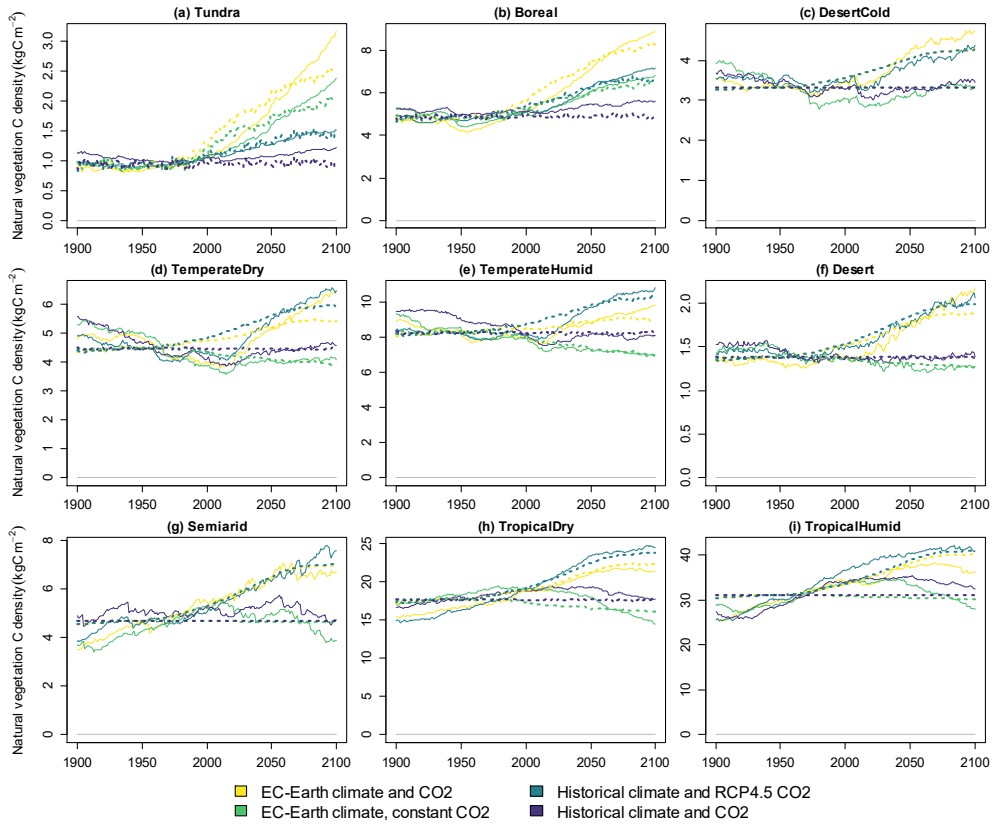

- EC-Earth climate and CO2    Historical climate and RCP4.5 CO2
- EC-Earth climate, constant CO2    Historical climate and CO2


**Figure 4. Carbon density in natural vegetation in the four simulated scenarios. Solid lines indicate LPJ-GUESS simulations and dashed lines indicate functions fitted to the results.**

The vegetation carbon stocks react strongly to climate change, and the magnitude of the effect depends on the biome. With a
constant climate and $CO_2$ concentration the densities remain relatively constant, as can be expected. An elevated $CO_2$ concentration increases the vegetation carbon density through the $CO_2$ fertilization effect. A warming climate increases carbon density in the cold biomes; but has negligible, or in some cases a small decreasing effect, on the other biomes. These effects are generally well in line with previous observations and model experiments regarding the global greening (Piao et al., 2020).



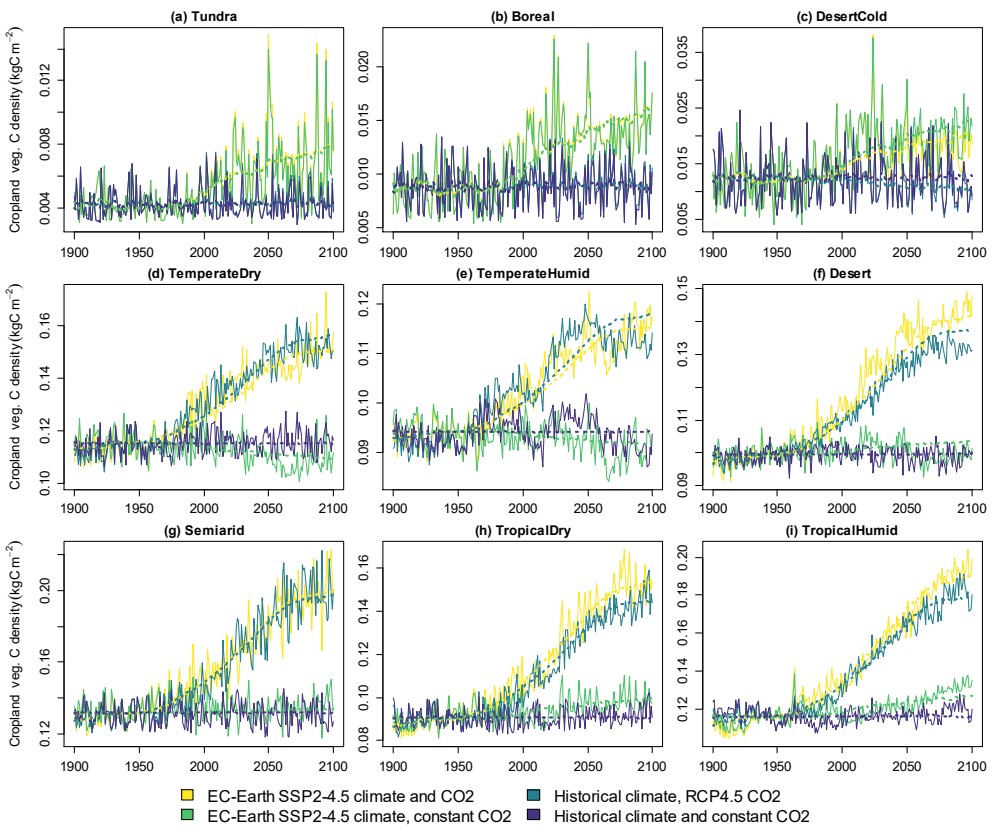


**Figure 5. Carbon density in cropland vegetation in the four simulated scenarios. Solid lines indicate LPJ-GUESS simulations and dashed lines indicate functions fitted to the results.**



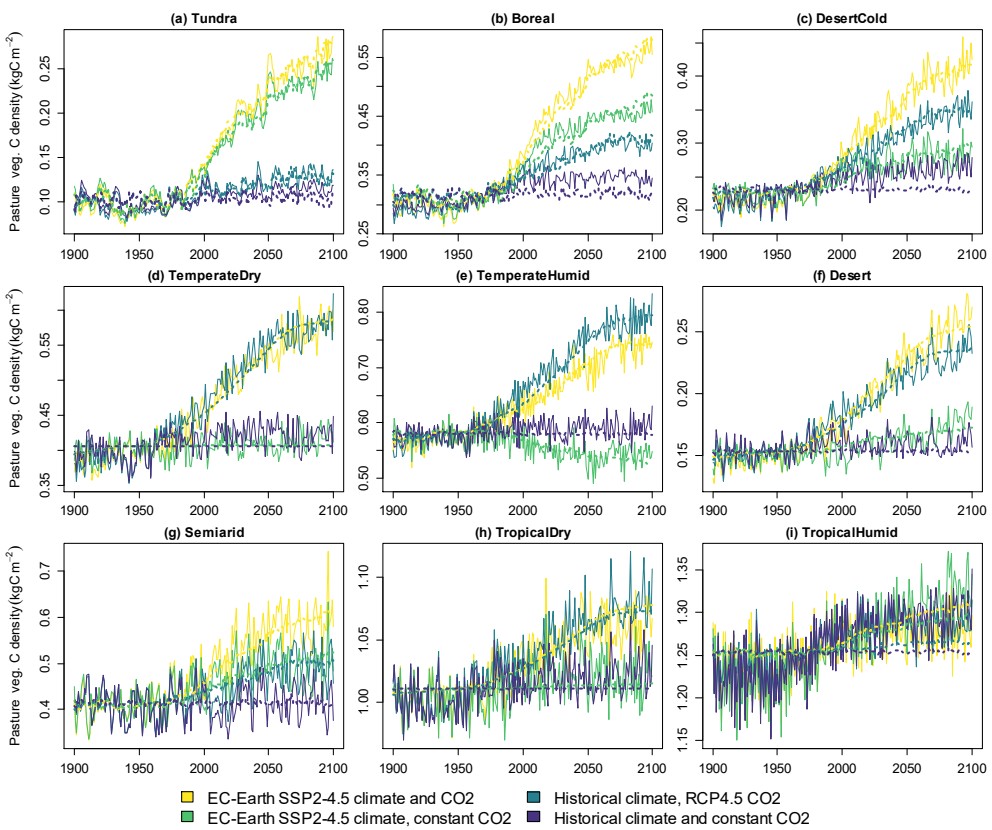

**Figure 6.** Carbon density in pasture vegetation in the four simulated scenarios. Solid lines indicate LPJ-GUESS simulations and dashed lines indicate functions fitted to the results.

### 3.6 Fits of litter and soil carbon stocks

The litter and soil carbon stocks of forests, croplands and pastures are presented in Appendix A. The fitted functions mostly compare well against the LPJ-GUESS simulations in the four climate scenarios described in Table 3. Relative differences between the fitted functions and the original LPJ-GUESS simulations are the largest for cropland carbon stocks. This is because the functions for cropland and pasture share the same parametrization, as they both contribute to the herbaceous litter and soil stocks in CLASH. Pastures contain more carbon than croplands and, therefore, have more weight when the litter and soil carbon dynamics functions are parametrized by minimizing the squared error between the LPJ-GUESS result and the fit. Hence, the fit is better for pastures than croplands. For the same reason, however, inaccuracy in depicting cropland litter and soils does not notably affect the overall accuracy of CLASH in emulating LPJ-GUESS simulations. That is, as cropland litter



and soil contain only a small part of the total carbon, inaccuracy in depicting these stocks does not notably affect the overall accuracy of representing the total carbon stocks.

### 3.7    Fits of crop yield

The yield of the average crop is presented in Figure 7. As earlier, the colder biomes experience a notable increase in yields in
a warming climate. Also, $CO_2$ fertilization improves yields notably. The statistical fits capture these changes well. The desert biome contains a discontinuity in the LUH2 cropland areas between the historical period and scenario after 2015, which is not captured by the fit.

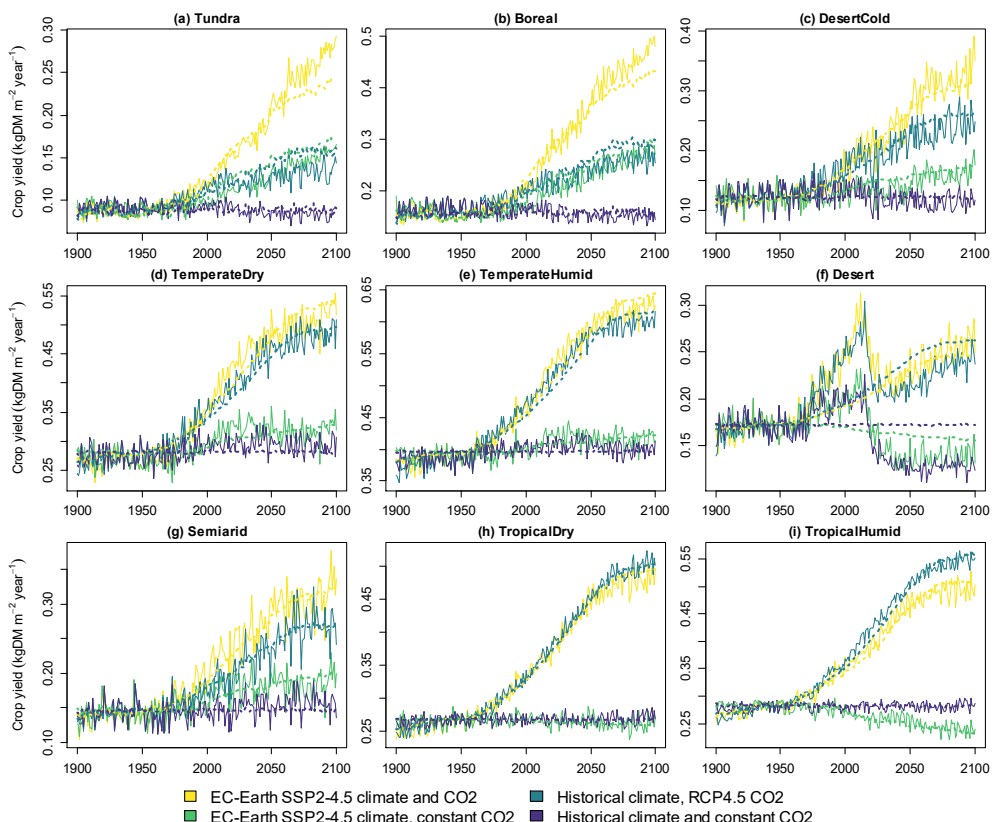

**Figure 7. Crop yields in the four simulated scenarios. Solid lines indicate LPJ-GUESS simulations and dashed lines indicate functions**
**fitted to the results.**





The $CO_2$ fertilization effect is notably strong on crop yields. In the temperate and tropical biomes, which comprise approximately 75% of cropland area in 2020, the average crop yields increase respectively by 38% and 57% between 2000 and 2100 due to the $CO_2$ from the RCP4.5 concentration pathway. This is higher than the approximately 15% to 30% increase

in a multi-model mean for four staple crops for the same concentration difference reported in (Franke et al., 2020). However, in that study LPJ-GUESS produced the highest response to elevated $CO_2$ among the compared models, and our results are in line with these earlier LPJ-GUESS results. The model, along with LPJmL, has also earlier been observed to produce stronger $CO_2$ fertilization effect than other DVGMs (Müller et al., 2015). Hence, a high response to changes in the atmospheric $CO_2$ concentration is a property of LPJ-GUESS, which CLASH correctly replicates.

**4    Validation**

We validated CLASH by comparing its results to those from LPJ-GUESS in the SSP2-4.5 scenario. Both models used the LUH2 land-use patterns (Hurtt et al., 2020). CLASH parametrization based on EC-Earth3 was compared to LPJ-GUESS driven with the EC-Earth3 SSP2-4.5 scenario; and similar comparisons were done using MPI and CanESM climate scenarios. While the LUH scenario determines land area allocation between different uses, it does not specify how secondary forests are

managed. Hence, to allow comparisons between the models, we applied forest management assumptions that lead to similar management. In the LPJ-GUESS LUH scenario, all forests were modelled without harvests. This behavior was emulated in CLASH with an exogenous objective to maximize terrestrial carbon stocks in 2100, which effectively minimizes harvests.

The results of the validation experiment are portrayed in Figure 8 with the EC-Earth3 climate scenarios. The models' results mostly align well with each other. The relative difference in the total terrestrial biosphere carbon stock ranges from 0.7% to

3% over the modelled examined timeframe. The differences are larger for specific carbon stocks in certain biomes, such as vegetation in the tropical biomes.

Three main reasons can potentially explain differences in results between the two models. (1) The resolution of the aggregate biome-level representation in CLASH is coarser than compared to the 2°×2° grid applied in LPJ-GUESS. (2) Inaccuracies in the fitted functions describing the processes in CLASH could cause the results to differ. (3) The areas of different land uses in

the LUH2 dataset are interpreted slightly differently in the two models, which also affect carbon stocks.

The main differences observed in Figure 8 can be attributed to differences in resolution (1) and differences in the interpretation of LUH2 data (3). The first of these inevitable, as CLASH describes the average growth, yield and carbon stocks over much larger areas than LPJ-GUESS. This was identified as the primary reason for the difference in vegetation carbon stock in the two tropical biomes. The problem could be remedied by applying a lower level of aggregation for the biomes, but this would

increase the computational weight of the model.

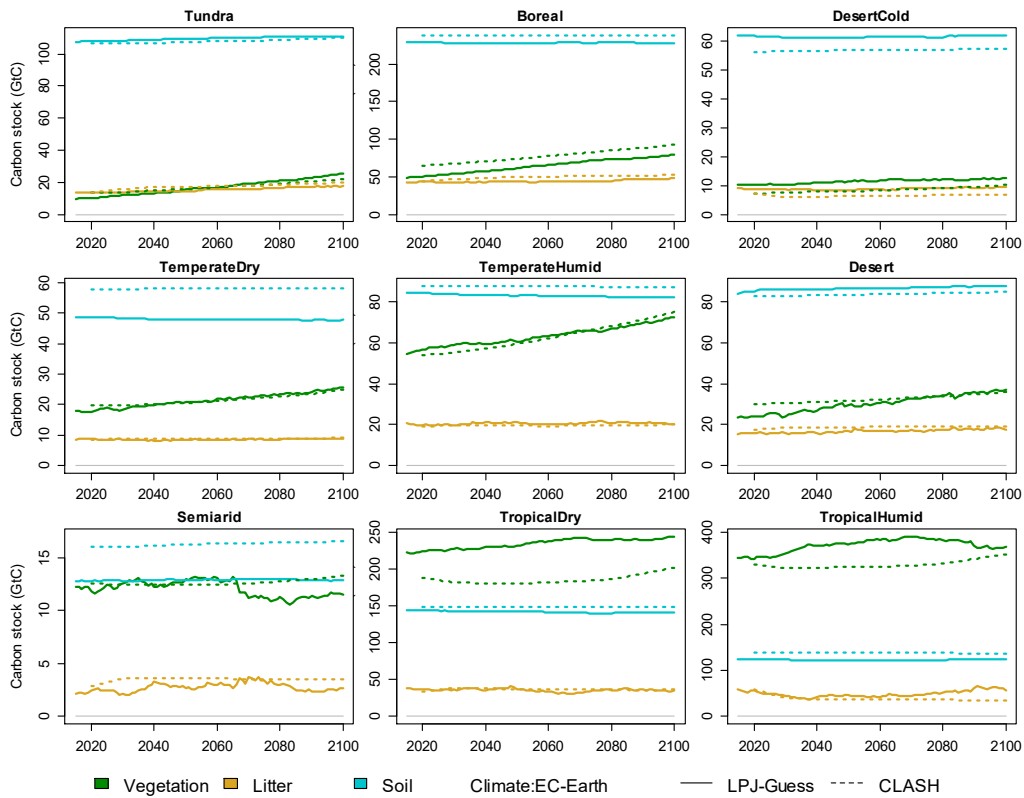

**Figure 8. Validation of CLASH carbon stocks against LPJ-GUESS in the SSP2-4.5. scenario, separately for vegetation, litter and soil carbon in each biome. Solid lines indicate LPJ-GUESS results, dashes CLASH results.**


Inaccuracies in the fitted functions (2) as a source of error could be reduced by trying to find functional forms that better emulate the LPJ-GUESS results. In general, however, the chosen functions and estimated parameters seem to replicate the LPJ-GUESS results relatively well. Cropland litter and soil carbon dynamics are an exception, but as these carbon stocks are minor compared to those of other land-uses, their effect is small in the big picture.

Differences in the interpretation of the LUH2 dataset (3) imply that land allocation within the biomes differs slightly between the models. The original LUH2 data is at 0.33°×0.33° resolution. The area of land allocated to each use in each biome in CLASH is calculated directly from this data. However, for LPJ-GUESS, the data is re-gridded, and a 2°×2° grid is applied in the LPJ-GUESS simulations. Different gridding leads to differences in biomes' land-use allocations between the models. This difference particularly explains the differences in soil carbon stocks in the Temperate Dry and Semiarid biomes, which have

roughly 20% more pasture area in CLASH.



Validation of CLASH fitted to the climate scenarios from CanESM and MPI models are presented in Appendix A (Figure A9 and Figure A10). These figures are qualitatively very similar to Figure 8, which indicates that the different parametrizations of CLASH can emulate well the LPJ-GUESS results driven by climate scenarios from different ESMs. However, it is worth noting that the choice of climate model choice notably affects the level of certain carbon stocks in the LPJ-GUESS results.

This is illustrated in Figure 9 with the three alternative parametrizations of CLASH. Whereas, across parametrizations, the carbon stocks are similar for most biomes, there is particularly notable difference in the Tundra biome, where the use of the CanESM climate scenario produces significantly larger carbon stocks for all three carbon pools. This can be explained by the higher temperature in the CanESM scenario compared to the other two climate scenarios. The higher temperature means greater vegetation growth, more litter input and hence, larger soil carbon stocks.


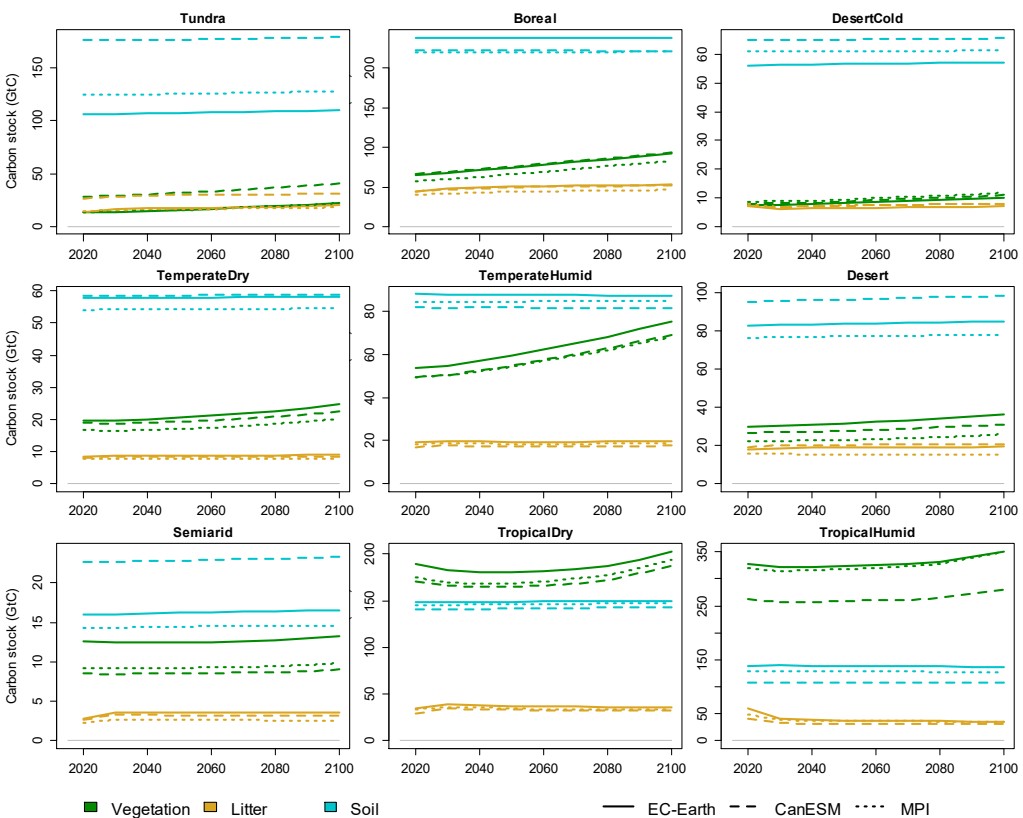

**Figure 9. Vegetation, litter and soil carbon stocks in a SSP2-4.5. scenario, modelled by CLASH parametrized against LPJ-GUESS results driven climate scenarios from EC-Earth (solid lines), CanESM (dashed lines) and MPI (dotted lines) models.**





## 5   Analysing trade-offs between carbon storage and production

To demonstrate the behaviour and possible use cases of CLASH, we explored how varying the future demand for different land-use products affects the global terrestrial carbon sink during the 21st century. In the example, CLASH is run subject to an exogenously given objective: maximize the terrestrial carbon stock in 2100 while satisfying an exogenously given demand scenario for agriculture and forestry products.

There is a trade-off between storing carbon in the biosphere and producing land-intensive products: more production usually
implies less storage (Erb et al., 2018). Examining this trade-off can help understand the physical limitations of the land-use sector's contribution towards mitigating climate change; e.g. to reach the Paris Agreement's 1.5°C target (Roe et al., 2019). Given these physical limitations in land-use, this trade-off can be seen as a production possibility frontier between carbon storage and the supply of land-use products (Pingoud et al., 2018). Here, we use CLASH to analyse this problem as an example of how the model can be utilized in practice.

Herein, CLASH is allowed to freely allocate land between cropland, pastures, and secondary forest across all biomes. The areas of other land-uses develop according to the SSP2-4.5 LUH scenario (Hurtt et al., 2020). Land-use products are represented by four aggregate categories: food crops (for direct human consumption), animal products (meat, milk, and eggs), wood products (timber and pulp wood), and bioenergy (energy crops and energy wood).

The model is first solved in a baseline scenario, in which the demand for each product category follows the SSP2-4.5 scenario
from 2020 to 2100 (Riahi et al., 2017), denoted as $D_{BL}(t)$. Then, we vary the demanded quantity of each product category at a time from the baseline, so that the demand $D(t)$ deviates gradually from the baseline until a variation $m$ of ±10% or ±50% is reached by 2100:

$$D(t) = D_{BL}(t) \cdot \left(1 + m \cdot \frac{t - 2020}{2100 - 2020}\right). \tag{12}$$

Hereafter, the demand scenarios are referred to as very low (-50%), low (-10%), high (+10%) and very high (+50%) demand. Figure 10 shows how changes in the demand for each product category affect the global carbon storage and net $CO_2$ uptake of
terrestrial ecosystems. The stocks respond almost linearly to changes in the demanded quantities. Altogether, the increase in global carbon stocks between 2020 and 2100 ranges from ±0 to +360 GtC across the demand scenarios, or -100% to +64% relative to the baseline increase of 220 GtC. Variation in animal product demand is responsible for the extremes of the range; and when measured per tonne of product, affects the average $CO_2$ uptake 9–10 times as strongly as variation in the crop or bioenergy demand, and 12 times as strongly as variation in wood demand.



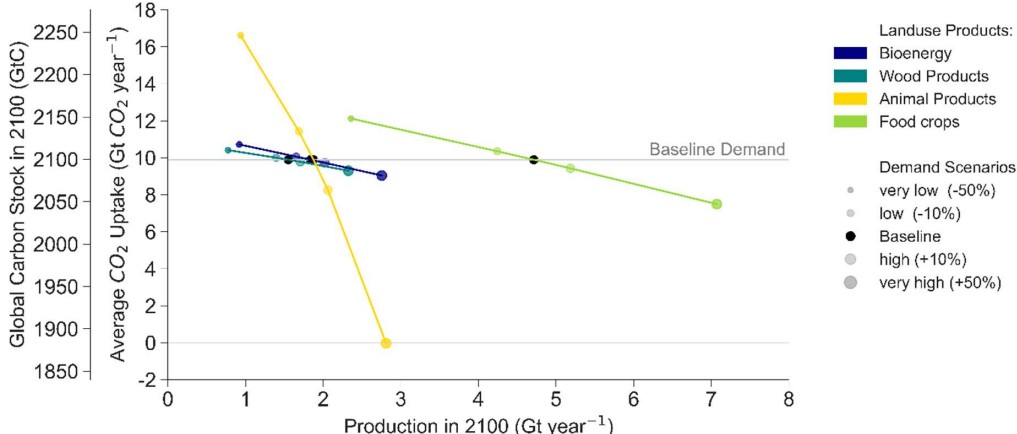


**Figure 10. The production-possibility frontier between carbon storage (y-axis) and the production of different land use products (x-axis) for food crops, animal products (meat, milk, and eggs), wood products (logs and pulp wood) and bioenergy (crops and logging waste). The carbon storage in the baseline demand scenario is indicated with a horizontal grey line. The demand variations are represented with point size and color: the point size indicates the relative deviation in 2100 from the baseline demand for the product**
**category indicated with the color. The slope of the line connecting the points indicates the sensitivity of global carbon stock to a change in demand per tonne of product.**

Demand variations cause land use conversions between cropland, pastures and secondary forests (Figure 11a). High demands for crops, animal products, or bioenergy are satisfied by converting additional secondary forest to agricultural land; while in the low demand scenarios, unused agricultural land can be converted to secondary forests to increase the global carbon stock.

Croplands are mostly allocated to temperate biomes and pastures to dry biomes. Cropland and pastures together make up 43–47% of total land area in 2050 and 34–65% in 2100 (Figure 11a). The loss of primary ecosystems from 2020 to 2100 amounts to 14.6 – 20.2% The effect of animal product demand on the loss of primary ecosystems is 9 times stronger than that of wood demand, 44 times stronger than that of bioenergy demand, and 105 times stronger than that of food crop demand.

The results suggest that – when assessed purely in terms of biophysical properties – the Tropical Humid biome has a relative
advantage in storing carbon over producing crops and timber, whereas the opposite is true for the temperate biomes (Figure 11b and Figure 11c). Dry, arid and desert biomes have a relative advantage in supporting livestock.

Hence, in our illustrative case, the production of crops and timber largely shifts to the temperate humid biomes; pastures are mostly allocated to dry biomes, and the tropical humid biome is mainly used for carbon storage. Only some pasture is allocated to the temperate and tropical humid biomes, particularly in the case of very high animal product demand. As the temperate
biomes are responsible for a large part of crop and wood production, their carbon stocks decrease the most during the century (Figure 11b).

Wood demand is primarily satisfied by harvesting large parts of Temperate Humid biome's secondary forests during the first half of the century and gradually converting them to croplands. Due to land-use change and intensive wood harvests, the biome's mean age of secondary forests decreases from 84 years in 2020 to 11 years in 2100 (Figure 11c). The remaining

demand for wood is met by harvests in boreal biome, which otherwise has a relative advantage for storing carbon. Large-scale forestation takes place in Boreal agricultural land, and the Boreal region then serves as the main supplier for wood products during the second half of the century. Strong afforestation combined with moderate harvesting increases the biome's carbon stock by 61 GtC compared to 2020; while the mean forest age decreases from 101 years in 2020 to 92 years in 2100 (Figure 11c).

Notably, the maximization of global carbon storage subject to the global demand constraints leads to a strongly polarized land allocation between the biomes. Economic factors (such as production costs, trade policies, security of supply concerns, and the value of ecosystem services) are not considered in this optimization problem. Doing so would alter regional relative advantages between carbon storage and production, and lead to a different global land allocation.

The $CH_4$ and $N_2O$ emissions from agricultural activities reach 14–32 Gt $CO_2$-eq per year between 2090 and 2100 (Figure 11d),
of which 67–76% are from animal husbandry. In most scenarios, the terrestrial carbon sink in 2100 (-22 – +3 GtCO$_2$-eq year$^{-1}$) is not large enough to compensate fully for the agricultural non-$CO_2$ emissions and, therefore, land-use is a net source of GHG emissions in 2100. In the very high animal product demand scenario, terrestrial ecosystems already become a net emission source by 2060.

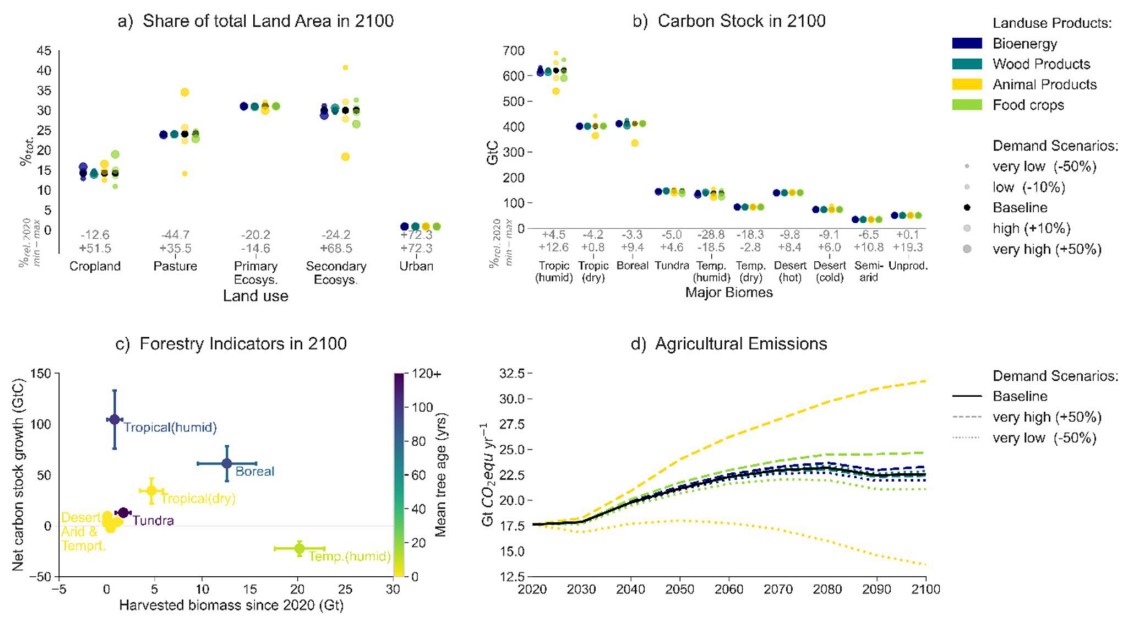


**Figure 11. a) Share of total land area in 2100 (%$_{tot.}$) and relative change since 2020 (%$_{rel.}$) per land use type in all scenarios. b) Total carbon stock in 2100 (GtC) and relative change (%$_{rel.}$) per biome in all scenarios. c) Total harvested biomass (Gt) and net carbon stock growth (GtC) for period 2020-2100, and area-weighted average tree age in 2100 (years), per biome averaged over the scenarios (error bars). d) Evolution of agricultural $CH_4$ and $N_2O$ emissions from crop cultivation, enteric fermentation, and manure**
**management (Gt $CO_2$-eq year$^{-1}$) for baseline and extreme demand scenarios. In panels a) and b), the point size indicates the relative**





**deviation in 2100 from the baseline demand for the product category indicated with the color. In panel d), the line type represents the relative variation in demand for the product category indicated with the color.**

Altogether, the large effect of animal products on climate, land-area, and ecosystems supports the view that reducing their consumption could be an effective means to mitigate climate change (Hayek et al., 2021; Jarmul et al., 2020). Earlier research

has also suggested that the relocation of croplands (Beyer et al., 2022) and increasing carbon storage in forests (Sohngen & Mendelsohn, 2003) are effective ways to mitigate climate change. All these effects can be identified in our illustrative analysis conducted using CLASH.

## 6    Conclusions

CLASH is a lightweight biophysical model that represents land use at an aggregate level of ten biomes, each divided into six

land use classes. Vegetation growth and ecosystem carbon dynamics respond to climate change in CLASH; and the model keeps track of terrestrial carbon stocks and, thereby, also of terrestrial $CO_2$ emissions and sinks. CLASH has been specifically designed to be hard-linked with IAMs. It can be incorporated into models formulated as linear or nonlinear programming problems, and run under intertemporal optimization. In this role, CLASH can be used to optimize global agriculture and forestry and their climatic impacts over a multi-decadal timescale. Hence, it can help evaluate the possible role that land-use

might have in mitigating climate change.

The role of land use in climate change mitigation has been extensively analyzed from various perspectives (e.g. Harper et al., 2018; Roe et al., 2019; Daioglou et al., 2019; Daigneault et al., 2022; Roebroek et al., 2023) and the topic's policy-relevance has recently increased due to the grown interest in maintaining and enhancing land-based carbon sinks (Griscom et al., 2017; Rockström et al., 2021). Our demonstration of CLASH in section 5 highlights the model's capacity to depict several well-

known mechanisms through which land-use can contribute to climate change mitigation, including reducing the consumption of animal products (Hayek et al., 2021; Jarmul et al., 2020), relocating croplands (Beyer et al., 2022) and increasing carbon storage in forests (Sohngen & Mendelsohn, 2003).

When embedded in an IAM, CLASH depicts the biophysical aspects of land-use, whereas the surrounding IAM provides the motivation for *how* the land should be used and managed, and describes the costs and constraints of land-use. The objective of

optimization-based IAMs is typically to either maximize welfare from consumption, or to minimize costs from satisfying predetermined demand (Keppo et al., 2021), possibly combined with climate targets or other policies. To ensure compatibility with a specific IAM, CLASH could be easily re-parametrized to alternative geographic resolutions and timesteps, instead of the 10 biomes and 10 years used here. It would also be possible to use CLASH as a pure simulation model, that is, without any optimization problem. However, this might be impractical due to the number of free variables and the equation structure of the

model. For this reason, we specified an external optimization problem of carbon stock maximization in section 5.

Due to its simplicity, CLASH cannot match the accuracy or detail of sectoral models, which have been soft-linked with IAMs (e.g. Fricko et al., 2017; Favero and Mendelsohn, 2014). CLASH's relative advantages are its light computational burden and





broad scope. It can be hard-linked to IAMs and run under intertemporal optimization to provide a comprehensive depiction of global land-use, terrestrial carbon stocks, and their bi-directional interaction with the climate. We believe this approach can be

helpful in examining the optimal role of land-use in mitigating climate change, providing food and biogenic raw-materials for the economy, and in conserving primary ecosystems.

### Appendix A

This appendix provides technical information on the model and additional figures to support the main text. The figures portray annual mean temperature and precipitation for each biome in the EC-Earth3 scenarios, the fits for litter and soil carbon in

forests, pastures and croplands, and validations using the parametrizations of CLASHS based on CanESM and MPI models.

### Model structure and use

CLASH is written in GAMS. The land-use module files are located in the main directory and the ecological module in its own subdirectory, which further contains separate subdirectories for the three parametrizations of CLASH based on the different ESMs' climate scenarios.

The ecological module produces GDX files that contain the parameterization of the land-use module. The input parameters are read from text files produced by the calibration scripts written in R. These text files are provided as a part of CLASH, and running the R scripts would be necessary only if the CLASH would be recalibrated for new ESM scenarios, regional definitions, or functional forms. The ecological module also reads a scenario for temperature change and atmospheric $CO_2$ concentration. This file can be provided externally, based on the ESM scenarios, or it can be produced by the IAM to which

CLASH is embedded.

For using the model within an IAM, the CLASH_Core.gms file needs to be included in the IAM source code. The IAM needs to provide the set *time* (alias *t*), subsets *tfirst* and *tlast*, the objective function and the model and solve statements. To run CLASH as a stand-alone model, as was done in this manuscript, a main file needs to provide these sets, the objective function and the MODEL and SOLVE statements. The CLASH code repository contains CLASH_Wrapper.gms file, which can be used

for this purpose as the main file.

### Specifications of climate the scenario experiments used in LPJ-GUESS

**Table A1. Climate scenario experiments used in LPJ-GUESS.**

| Model | Experiments | Variant | Original resolution (gridpoints) | DOI | Citation |
|-------|-------------|---------|-----------------------------------|-----|----------|
|       |             |         |                                   |     |          |



| EC-Earth | Historical, ssp245 | r1i1p1f1 | 512 x 256 longitude/latitude | https://doi.org/10.22033/ESGF/CMIP6.4700, https://doi.org/10.22033/ESGF/CMIP6.4880 | EC-Earth Consortium, 2019a,b |
|---|---|---|---|---|---|
| CanESM5 | Historical, ssp245 | r10i1p1f1 | 128 x 64 longitude/latitude | https://doi.org/10.22033/ESGF/CMIP6.3610, https://doi.org/10.22033/ESGF/CMIP6.3685 | Swart et al, 2019a,b |
| MPI-ESM | Historical, ssp245 | r1i1p1f1 | 192 x 96 longitude/latitude | https://doi.org/10.22033/ESGF/CMIP6.6595, https://doi.org/10.22033/ESGF/CMIP6.6693 | Wieners et al, 2019a,b |

EC-Earth: historical **r1i1p1f1**

EC-Earth Consortium (EC-Earth) **(2019)**. *EC-Earth-Consortium EC-Earth3 model output prepared for CMIP6 CMIP historical*. Version *20200310*. Earth System Grid Federation. https://doi.org/10.22033/ESGF/CMIP6.4700

EC-Earth s**sp245 r1i1p1f1f r1i1p1f1**

EC-Earth Consortium (EC-Earth) **(2019)**. *EC-Earth-Consortium EC-Earth3 model output prepared for CMIP6 ScenarioMIP*

*ssp245*. Version *20200310*. Earth System Grid Federation. https://doi.org/10.22033/ESGF/CMIP6.4880

Original grid TL255, linearly reduced Gaussian grid equivalent to 512 x 256 longitude/latitude

CanESM5 **historical r10i1p1f1**

Swart, N.C., Cole, J.N.S., Kharin, V.V., Lazare, M., Scinocca, J.F., Gillett, N.P., Anstey, J., Arora, V., Christian, J.R., Jiao,

Y., Lee, W.G., Majaess, F., Saenko, O.A., Seiler, C., Seinen, C., Shao, A., Solheim, L., von Salzen, K., Yang, D., Winter, B., Sigmond, M. **(2019)**. *CCCma CanESM5 model output prepared for CMIP6 CMIP historical*. Version 20190429. Earth System Grid Federation. https://doi.org/10.22033/ESGF/CMIP6.3610

**CanESM5 ssp245 r10i1p1f1**

Swart, N.C., Cole, J.N.S., Kharin, V.V., Lazare, M., Scinocca, J.F., Gillett, N.P., Anstey, J., Arora, V., Christian, J.R., Jiao, Y., Lee, W.G., Majaess, F., Saenko, O.A., Seiler, C., Seinen, C., Shao, A., Solheim, L., von Salzen, K., Yang, D., Winter, B., Sigmond, M. **(2019)**. *CCCma CanESM5 model output prepared for CMIP6 ScenarioMIP ssp245*. Version *20190429*. Earth System Grid Federation. https://doi.org/10.22033/ESGF/CMIP6.3685

Original grid: T63 Linear Gaussian Grid; 128 x 64 longitude/latitude


MPI-ESM historical **r1i1p1f1**

Wieners, Karl-Hermann; Giorgetta, Marco; Jungclaus, Johann; Reick, Christian; Esch, Monika; Bittner, Matthias; Legutke, Stephanie; Schupfner, Martin; Wachsmann, Fabian; Gayler, Veronika; Haak, Helmuth; de Vrese, Philipp; Raddatz, Thomas;



Mauritsen, Thorsten; von Storch, Jin-Song; Behrens, Jörg; Brovkin, Victor; Claussen, Martin; Crueger, Traute; Fast, Irina;
Fiedler, Stephanie; Hagemann, Stefan; Hohenegger, Cathy; Jahns, Thomas; Kloster, Silvia; Kinne, Stefan; Lasslop, Gitta;
Kornblueh, Luis; Marotzke, Jochem; Matei, Daniela; Meraner, Katharina; Mikolajewicz, Uwe; Modali, Kameswarrao; Müller,
Wolfgang; Nabel, Julia; Notz, Dirk; Peters-von Gehlen, Karsten; Pincus, Robert; Pohlmann, Holger; Pongratz, Julia; Rast,
Sebastian; Schmidt, Hauke; Schnur, Reiner; Schulzweida, Uwe; Six, Katharina; Stevens, Bjorn; Voigt, Aiko; Roeckner, Erich
**(2019)**. *MPI-M MPI-ESM1.2-LR model output prepared for CMIP6 CMIP historical*. Version *20190710*. Earth System Grid
Federation. https://doi.org/10.22033/ESGF/CMIP6.6595

MPI-ESM **ssp245 r1i1p1f1**

Wieners, Karl-Hermann; Giorgetta, Marco; Jungclaus, Johann; Reick, Christian; Esch, Monika; Bittner, Matthias; Gayler,
Veronika; Haak, Helmuth; de Vrese, Philipp; Raddatz, Thomas; Mauritsen, Thorsten; von Storch, Jin-Song; Behrens, Jörg;
Brovkin, Victor; Claussen, Martin; Crueger, Traute; Fast, Irina; Fiedler, Stephanie; Hagemann, Stefan; Hohenegger, Cathy;
Jahns, Thomas; Kloster, Silvia; Kinne, Stefan; Lasslop, Gitta; Kornblueh, Luis; Marotzke, Jochem; Matei, Daniela; Meraner,
Katharina; Mikolajewicz, Uwe; Modali, Kameswarrao; Müller, Wolfgang; Nabel, Julia; Notz, Dirk; Peters-von Gehlen,
Karsten; Pincus, Robert; Pohlmann, Holger; Pongratz, Julia; Rast, Sebastian; Schmidt, Hauke; Schnur, Reiner; Schulzweida,
Uwe; Six, Katharina; Stevens, Bjorn; Voigt, Aiko; Roeckner, Erich **(2019)**. *MPI-M MPI-ESM1.2-LR model output prepared*
*for   CMIP6   ScenarioMIP   ssp245*.   Version   *20190710*.   Earth   System   Grid   Federation.
https://doi.org/10.22033/ESGF/CMIP6.6693

Original resolution spectral T63; 192 x 96 longitude/latitude



**Global mean temperature and precipitation**

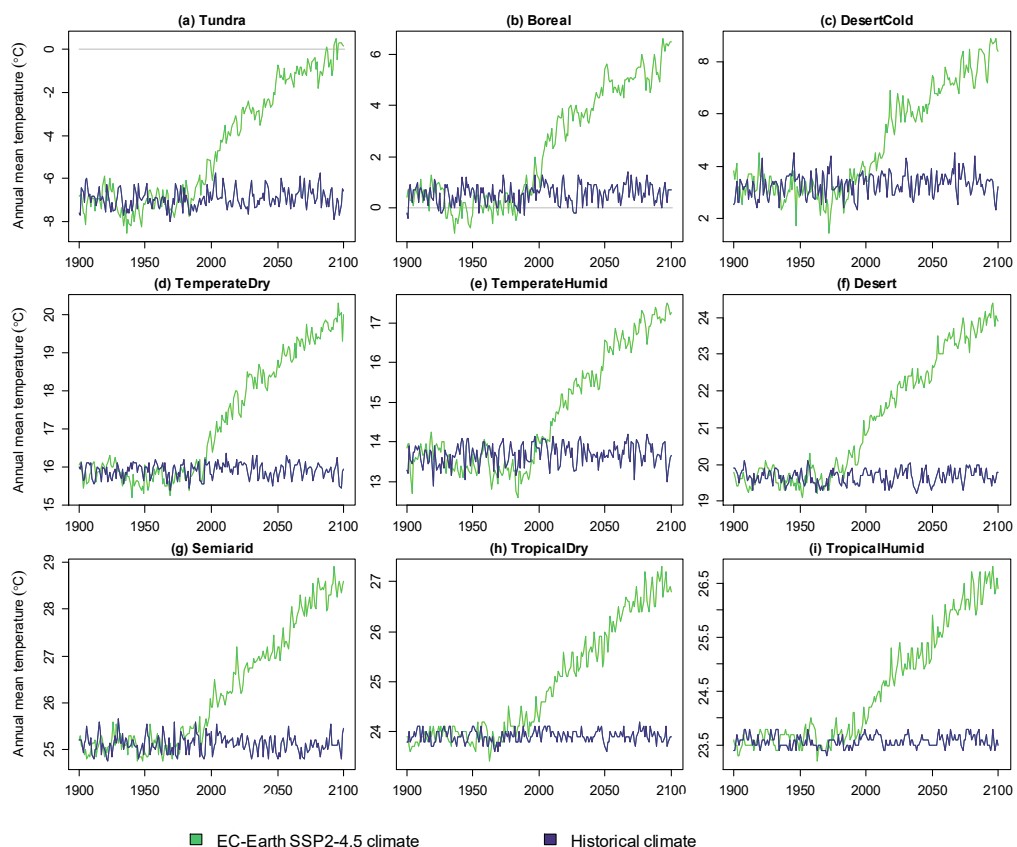

**Figure A1. Mean temperature in the nine main biomes with historical climate and EC-Earth SSP2-4.5 scenario.**



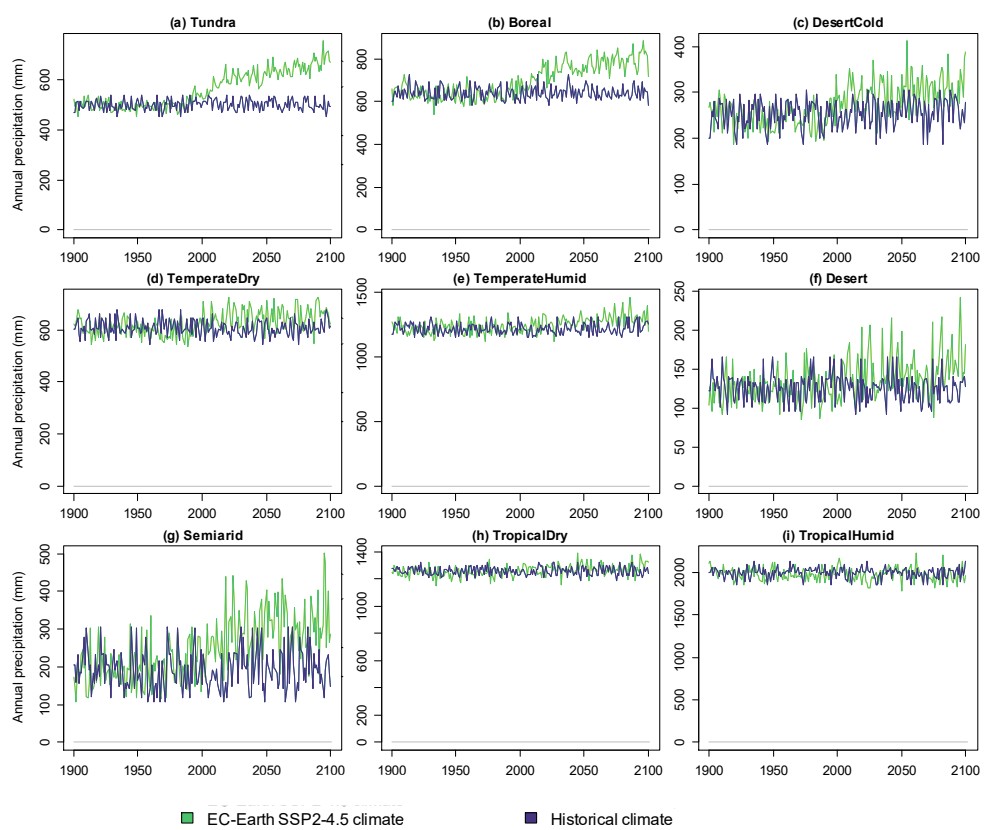

**Figure A2. Annual precipitation in the nine main biomes with historical climate and EC-Earth SSP2-4.5 scenario.**



**Fits of litter and soil carbon stocks**

The following figures portray the litter and soil carbon stocks simulated with LPJ-Guess, and the simulations with fitted functions to the results. These results are discussed in section 3.6.

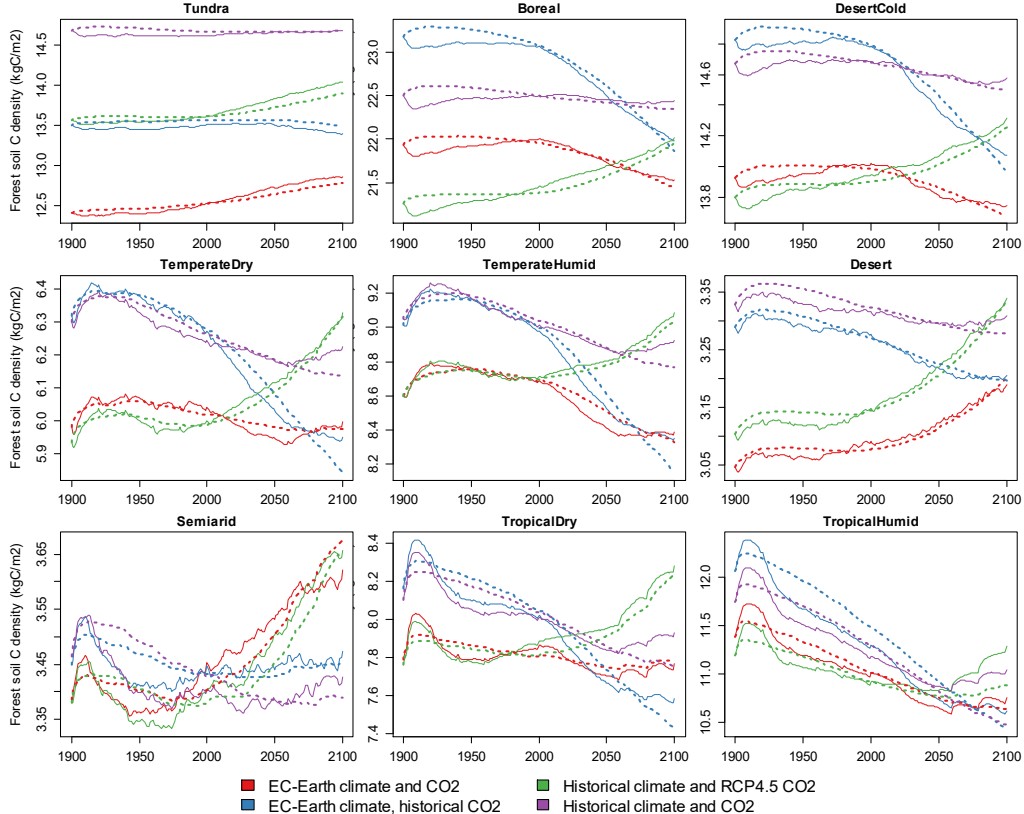

**Figure A3 Carbon density in forest soil in the four simulated scenarios. Solid lines indicate LPJ-GUESS simulations and dashed lines indicate functions fitted to the results.**




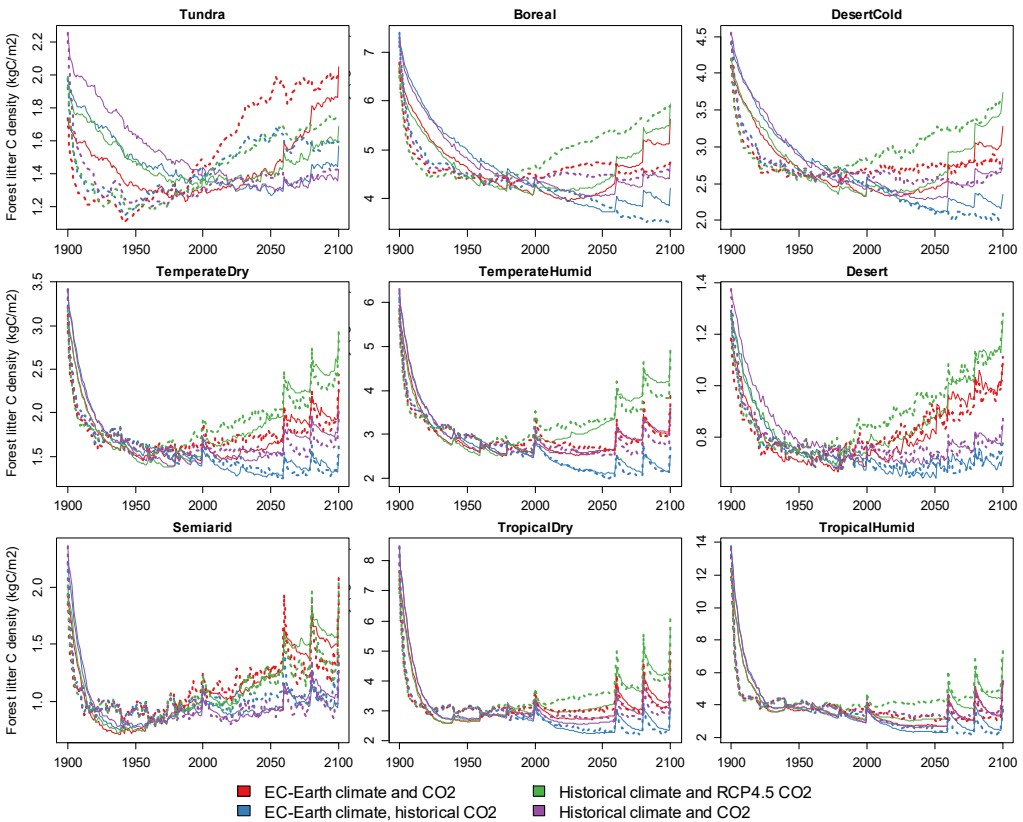

**Figure A4 Carbon density in forest litter in the four simulated scenarios. Solid lines indicate LPJ-GUESS simulations and dashed lines indicate functions fitted to the results.**



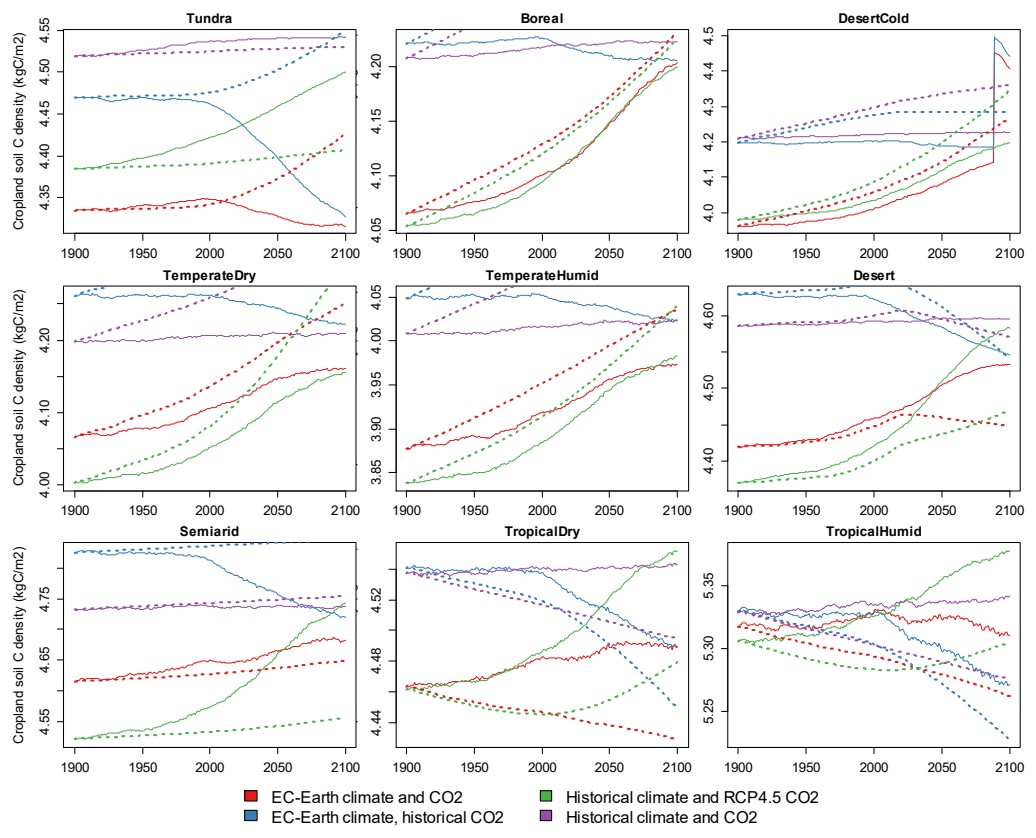


**Figure A5 Carbon density in cropland soil in the four simulated scenarios. Solid lines indicate LPJ-GUESS simulations and dashed lines indicate functions fitted to the results.**





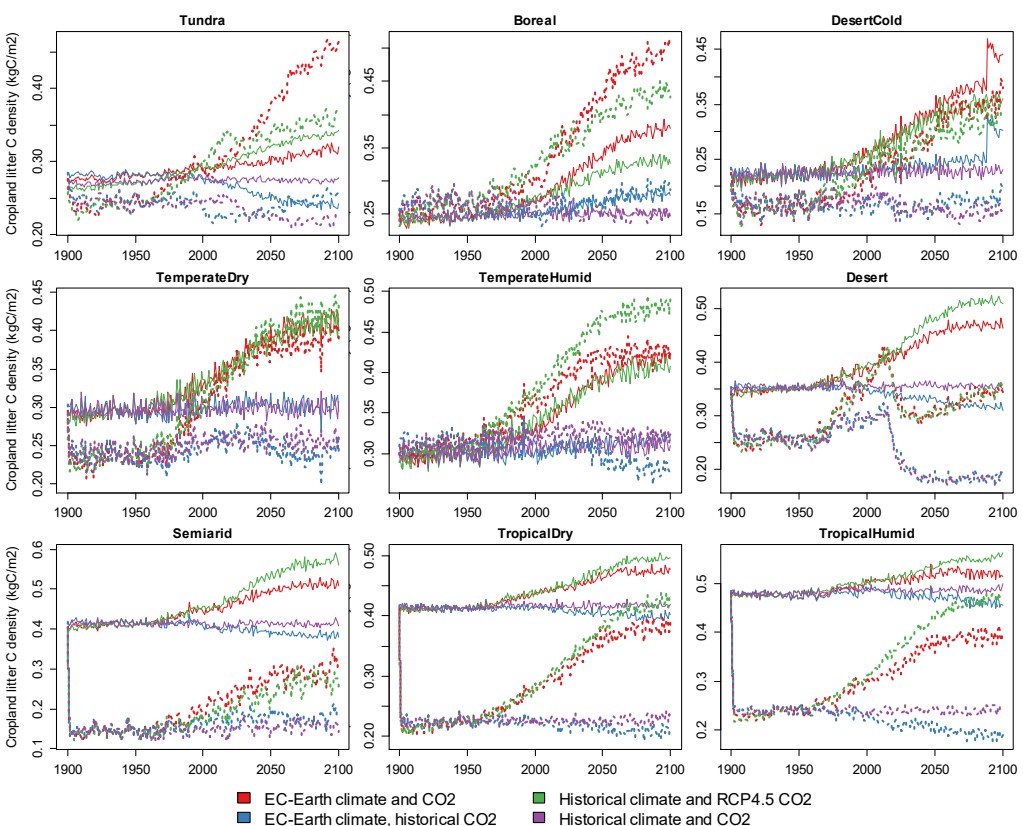

**Figure A6 Carbon density in cropland litter in the four simulated scenarios. Solid lines indicate LPJ-GUESS simulations and dashed lines indicate functions fitted to the results.**





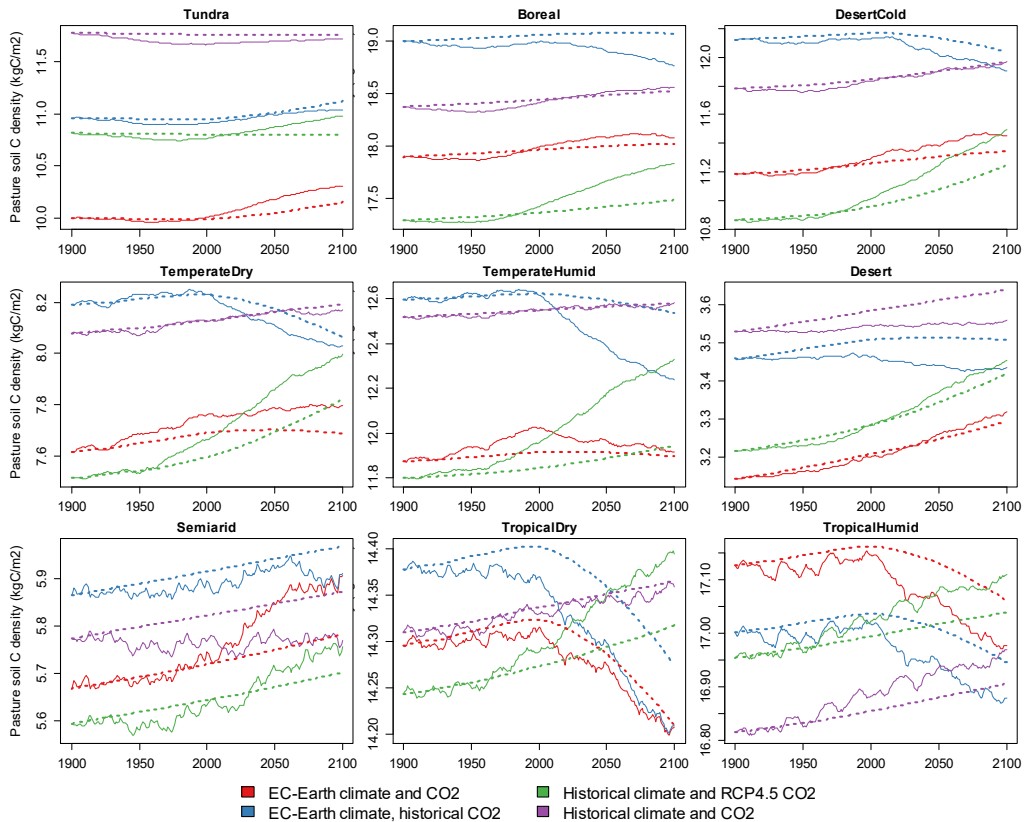

**Figure A7 Carbon density in pasture soil in the four simulated scenarios. Solid lines indicate LPJ-GUESS simulations and dashed lines indicate functions fitted to the results.**






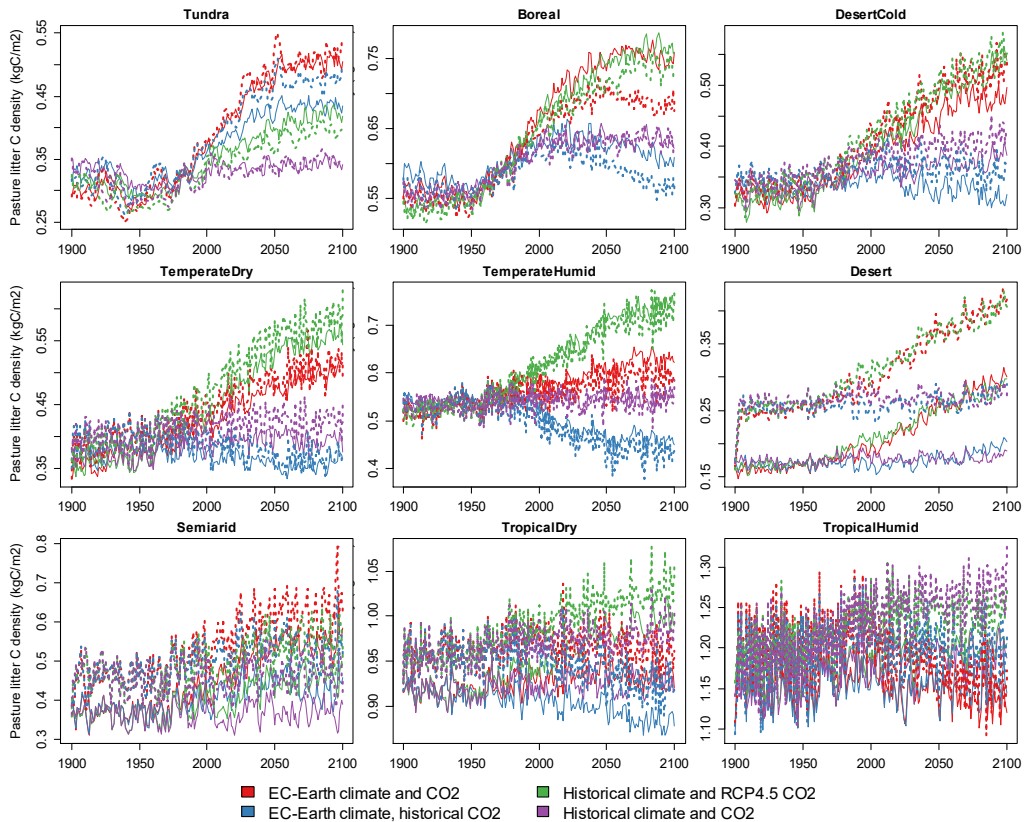

**Figure A8 Carbon density in pasture litter in the four simulated scenarios. Solid lines indicate LPJ-GUESS simulations and dashed lines indicate functions fitted to the results.**


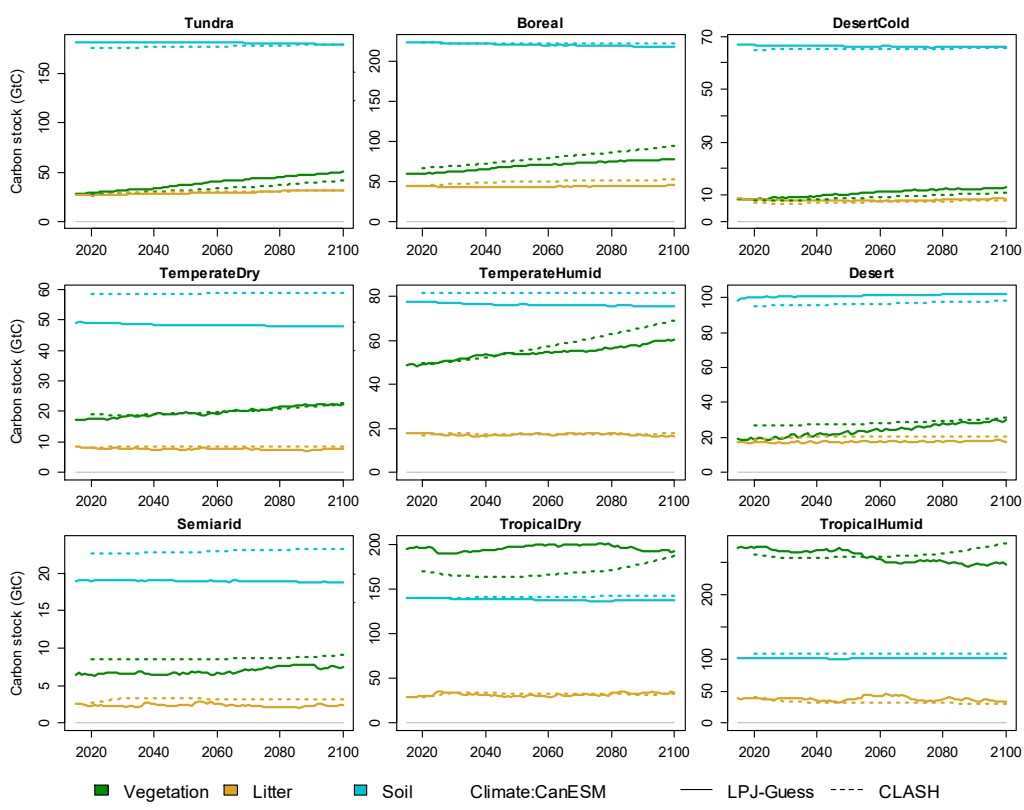

**Figure A9 Validation of CLASH carbon stocks against LPJ-GUESS in the SSP2-4.5 scenario calculated with the CanESM model; separately for vegetation, litter and soil carbon in each biome. Solid lines indicate LPJ-GUESS results, dashes CLASH results.**


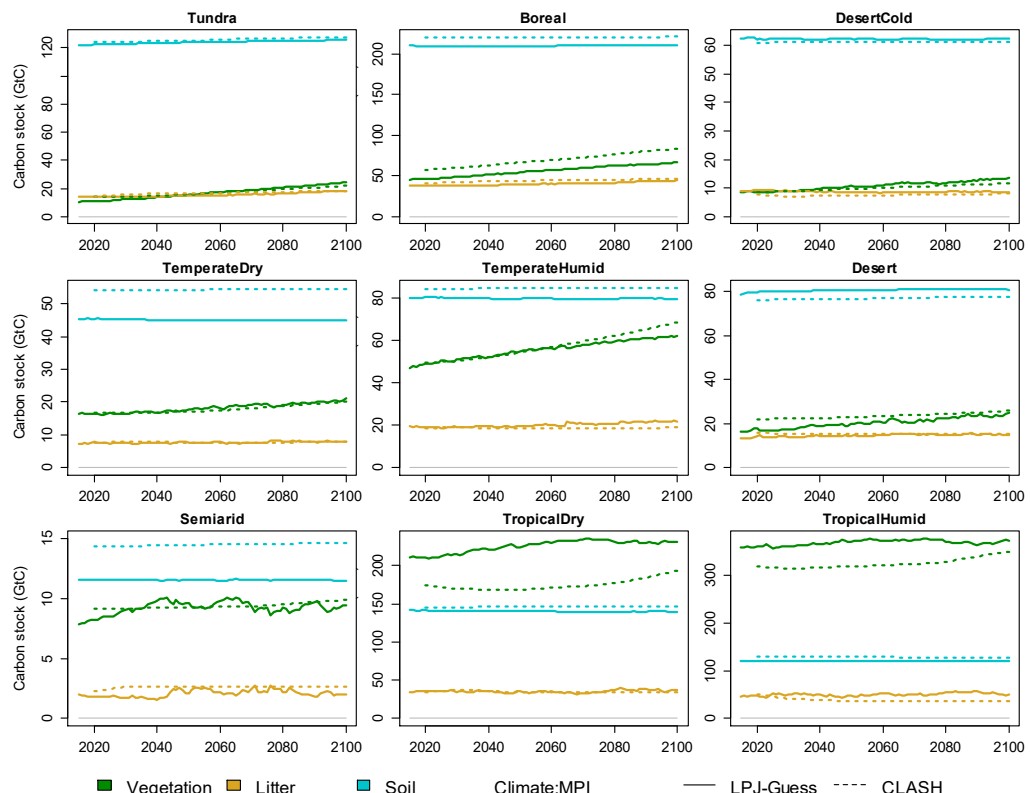

**Figure A10 Validation of CLASH carbon stocks against LPJ-GUESS in the SSP2-4.5 scenario calculated with the MPI ESM model; separately for vegetation, litter and soil carbon in each biome. Solid lines indicate LPJ-GUESS results, dashes CLASH results.**


*Code availability*. The current version of the GAMS code for CLASH is available at: https://github.com/tekholm/CLASH under the MIT License. The version used in this paper is archived on Zenodo (DOI:10.5281/zenodo.8271846), as are LPJ-GUESS results (DOI: 10.5281/zenodo.8272853) and R scripts (DOI:10.5281/zenodo.8273074) to parametrize the model, and

the demonstration results and scripts to produce the plots from these results (DOI:10.5281/zenodo.8272900).

*Author contributions*. Ekholm and Rautiainen conceived the model. Thölix performed LPJ-GUESS simulations and analyzed the results. Ekholm carried out the statistical fitting of model equations to the LPJ-GUESS results. Ekholm, Rautiainen and Freistetter programmed the CLASH model. Freistetter performed the demonstration case runs and analyzed their results. All authors contributed to writing the paper.

*Competing interests*. The authors declare that they have no conflict of interest.



*Acknowledgements*. The research has been done with funding from the Academy of Finland in projects SuCCESs (decision number 341311) and OptiMit (decision number 331491).

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
