# Peer review of "CLASH - Climate-responsive Land Allocation model with carbon Storage and Harvests"

_Geoscientific Model Development, 2023_

## Author Response (AR1)

**Response to Reviewers – GMD-2023-146**

The authors would like to thank both Reviewers and the Editor for insightful comments, which were helpful in improving the manuscript. All suggested changes were made to the revised manuscript. Additionally, some minor improvements were made to the model during the review process, and because of that, we updated the demonstration case in section 5. We also made some textual improvements, with the intention of improving the readability of the manuscript.

Our response and an indication of how the manuscript was revised is provided below each Reviewer comment, using bullet points and italics. A version of the manuscript that highlights all changes is also provided.

**Reviewer #1**

**General remarks**

In this manuscript, the authors present the CLASH model. According to the authors, CLASH is "a biophysical land-use model that describes the allocation of land to different uses, forest growth, terrestrial carbon stocks, and the production of agricultural and forestry goods globally." CLASH is designed for hard-linking with IAMs but can also be run in standalone mode.

According to the authors, CLASH "fills a critical, vacant niche in the model ecosystem" (line 49) by combining features from three model types: Dynamic Global Vegetation Models (DGVMs), economic partial equilibrium models of the land-use sector, and forest sector models.

Indeed, there is currently no such model available that combines features from these three model types in a simple and lightweight model with intertemporal optimization for hard-linking with IAMs. Therefore, the concept of CLASH is innovative.

- *We are glad that the Reviewer finds the model innovative and sees the vacant niche that we wanted to fill with this model. We are also grateful for the questions and observations by the Reviwer, which helped us to improve the manuscript further. All suggested changes were made to the revised manuscript.*

The paper is very well structured and written. The language is fluent and precise.

- *Thank you. We are happy if the text was clear to read.*

In the abstract, the authors claim that CLASH "allows the inclusion of terrestrial carbon stocks, agriculture and forestry in global climate policy analyses."

This might be true from a technical point of view. But it seems that economic factors such as trade patterns and self-sufficiency ratios, protected land areas, and existing climate policies for the land-use sector are largely missing in CLASH (see my detailed comments below).

- *Thank you for noting this, and also for the related observations below. It is important to clarify the scope of the model, so that readers or potential users of the model are not confused about what the model represents and what falls outside its scope.*
- *The scope of CLASH is exactly as the Reviewer has understood. The aim is that CLASH portrays only biophysical factors, whereas all economic or societal factors, such as those listed above, should come from the IAM to which CLASH would be linked; or given externally, if run as stand-alone. These have now been clarified in the revised manuscript. Please see the responses to the detailed comments below for further details.*
- *The cited text from the abstract has been re-written as: "Connecting CLASH to an IAM would allow the consideration of terrestrial carbon stocks, agriculture and forestry in global climate policy analyses."*

The absence of these factors, which are highly relevant for near-term model realism, considerably limits the usefulness of CLASH for climate policy analyses. In my view, CLASH is rather a useful tool for idealized (biophysical) experiments but not for concrete climate policy analyses. This aspect should be clarified by the authors in a major revision of the manuscript.

- *We are happy to hear that the Reviewer finds CLASH as a useful tool for idealized biophysical experiments. But indeed, factors like markets and policies are highly relevant for modelling real-world land-use. As these are outside the scope of CLASH, it alone does not provide a scope that is sufficient for policy analyses. Instead, economic and policy factors should be provided through the linkage to an IAM; or e.g. as external constraints when running stand-alone.*
- *We have clarified this aspect now in the introduction: "In this role, CLASH represents the biophysical aspects of land-use, while the IAM needs to provide the rationale for why land is used and managed in a specific way, including economic, policy and other societal factors; and also how the climate changes over time. These factors are relevant for realistic modeling of land-use, but are outside the scope of CLASH."*

- *And similarly in the conclusions: "When integrated with an IAM, CLASH provides the annual production of land-use commodities (food and other biomass for energy and materials), the change in terrestrial CO2 stocks and GHG emissions from agriculture. The IAM should provide the demand for these products, and any costs, policies and other societal constraints that affect land-use. As such, CLASH would depict land-use from a biophysical perspective, whereas the IAM provides the motivation for how the land should be used and managed."*

**Detailed comments**

1.) It remains unclear which features from economic partial equilibrium models such as MAgPIE or GLOBIOM are represented in CLASH. The authors claim that "CLASH provides a comprehensive, climate-responsive depiction of global land-use" (line 68f). However, they also admit: "This comes at the expense of detail. Ecosystem characteristics, land-use competition, agriculture and forest management are described in less detail than in models focusing on each aspect individually" (line 69f).

It's fair that a model like CLASH that aims to integrate features from different modelling domains represents less details from each domain. However, the authors should clarify in the introduction which features typically included in economic partial equilibrium models are captured by CLASH, and which features are missing.

Key features of economic partial equilibrium models like MAgPIE or GLOBIOM include food demand, agricultural production, factor costs for agricultural production, dynamic land-use competition, yield increases, accounting for historic trade patterns and self-sufficiency ratios, and GHG emissions.

Partly, the above point is addressed in section 5 (lines 520ff): "Economic factors (such as production costs, trade policies, security of supply concerns, and the value of ecosystem services) are not considered in this optimization problem." However, if economic factors are not accounted for in CLASH, which features of economic partial equilibrium models are then included in CLASH?

- *Thank you for this observation. The model scope indeed needs to be communicated more clearly. As discussed above, CLASH covers only biophysical factors, i.e. vegetation growth, carbon stocks, crop production and livestock (but only from a biophysical point-of-view). As compared to model focused on agriculture, the are no separate crops (only an aggregated crop) or specific management practices (although irrigation and fertilization are accounted implicitly through LPJ-GUESS); while in comparison to forestry models, tree species are not*

*considered explicitly (but only implicitly, through LPJ-GUESS) and managed forests cannot be thinned, but only clear-cut. Economic and policy factors (e.g. costs, markets, prices, trade and self-sufficiency ratios) are outside its scope, but can be introduced externally or when linked to an IAM.*

- *We now write in the introduction: "In particular, CLASH covers the climate-responsive vegetation growth and portrayal of vegetation and soil carbon stocks from DGVMs, crop and livestock production from land-use models, age-structured forests and harvesting of wood from forest sectors models, and the possibility for optimizing land-use from both types of partial equilibrium models. What is left out is the product demand, markets and policies that drive land-use decisions in partial equilibrium models, and the detailed biophysical modeling of ecosystems in DGVMs. Also, agriculture and forest management are described in less detail and without detailed management options as in models focusing on each aspect individually."*

2.) Related to the above point on economic partial equilibrium models. In line 61f, the authors claim: "Both of these features [dynamic forest age classes and intertemporal optimization] are needed to enable the dynamic optimization of forest management when, e.g., studying global resource use or climate policy questions."

This sentence needs clarification. In my view, it cannot be generalized that intertemporal optimization is a precondition for the dynamic optimization of forest management. Instead, the choice of intertemporal optimization vs. recursive-dynamic optimization depends on the underlying research question. Intertemporal optimization tells us how land should be allocated to different uses over time to meet a certain goal such as carbon stock maximization. However, policy-making in the real world, especially in the land-use sector with its many small holder farmers, happens rather incrementally and not in an intertemporal fashion. In this sense, projections based on recursive-dynamic optimization are likely closer to what will happen in the future, compared to their intertemporal counterpart.

Moreover, the choice of intertemporal vs. recursive-dynamic optimization may also be informed by the properties of the sector under consideration. For instance, in the energy sector long-term planning is much more established compared to the land-use sector.

For improved context and clarity, I suggest that the authors revise the above statement, and add some sentences on the relation and choice between intertemporal vs. recursive-dynamic optimization.

- *Thank you for these insights. We agree completely that the different modeling paradigms can answer different questions, and thereby can represent different aspects of the real-world with different degrees of realism. Due to this, one is not inherently better than the other, but different approaches complement each other.*
- *Intertemporal optimization (with perfect foresight) might be a very idealized and optimistic approach, as all decision-makers are assumed to do long-term planning with perfect knowledge of future events. In this sense, recursive dynamics can maybe be considered a more realistic description to real-world decision-making. But with recursive dynamics, decision-making is reactive and disregards long-term foresight. This, we think, is not fully realistic either.*
- *But besides realism, the paradigm choice affects what can be answered with the model. Intertemporal optimization answers what would be an ideal long-term strategy. We think such analyses can be illuminating, even if the long-term strategy would be an unrealistic one due to the short-sightedness of real-world decision-makers. However, both approaches can remedy the 'missing realism' and emulate the behavior of the other approach through iterative procedures: running intertemporal optimization with myopia, or running recursive dynamics with appropriately set incentives that reach long-term targets, for example.*
- *Intertemporal optimization is computationally more challenging than recursive dynamics, as we discuss in the introduction. Exactly because of this, CLASH was designed to be simplified and computationally lightweight, so that it could be embedded with IAMs that employ intertemporal optimization. However, this does not prevent it from being integrated into an IAM using recursive dynamics.*
- *The difference between these two modeling paradigms might be less relevant for agriculture, but with long-rotation forestry it is more important. In forest economics, the classic Faustmann's problem of optimal rotation length, already from 1849, is defined as an intertemporal optimization problem. Thereby, we stand by our statement that dynamic forest age classes and intertemporal optimization are necessary to enable the long-term optimization of forest management. We think this is the only logical conclusion if 'optimization' is understood in a strict, mathematical sense.*
- *We have modified the referenced sentence in the following way: "Both features are needed to enable the full dynamic optimization of forest structure and management over long time horizons, e.g. when aiming for long-term climatic targets."*
- *We also added a short discussion based on the above response as a footnote: "Intertemporal optimization and recursive-dynamic optimization are two main ways of modelling optimal*

*actions over long time horizons. The main distinction is that an intertemporal problem finds optimal actions for the whole timeframe at once, whereas recursive-dynamics optimizes each timestep chronologically. The two approaches can provide complementary insights. Whereas intertemporal optimization can be seen as too 'idealized' as it assumes perfect foresight over the whole time horizon, recursive-dynamic can be seen as too myopic towards long-term developments. However, both approaches can simulate each other's behavior: intertemporal optimization through a myopic formulation, and recursive-dynamics through iterative procedures."*

3.)  It seems that real world trade patterns and self-sufficiency ratios are not considered at all in CLASH, which explains the following result (lines 520ff): "Notably, the maximization of global carbon storage subject to the global demand constraints leads to a strongly polarized land allocation between the biomes."

Disregarding trade and self-sufficiency ratios considerably limits the usefulness of CLASH for climate policy analysis. Without any near-term realism of land-use at regional / biome level, meaningful climate policy analysis is not possible. Therefore, the corresponding statements in the abstract (line 9) and in the conclusion (line 570) should be revised.

- *As discussed above, CLASH does not cover any societal factors, such as markets, trade or policies for self-sufficiency in itself. These could be given externally to the model, however, particularly if CLASH is connected to an IAM.*
- *In the demonstration that the Reviewer cites above, we deliberately wanted to keep the optimization problem simple, so that it is easy to see how and why the results arise from the optimization. The purpose of the demonstration was not to portray a realistic analysis for climate policy (for example: there is no linkage between crop and animal product demands), but to test the model in a simple setting that would be physically possible.*
- *The demonstration case is now introduced from this viewpoint: "In the example, CLASH is run subject to an exogenously given objective: maximize the terrestrial carbon stock in 2100 by allocating managed lands between the biomes, while satisfying an exogenously given demand scenario for agriculture and forestry products. As this problem setting does not portray the drivers determining real-world land-use decisions, the analysis should be seen as a simple demonstration of the model through scenarios that would still be physically possible."*

- *Line 9 in the abstract has been re-written as: "Connecting CLASH to an IAM would allow the consideration of terrestrial carbon stocks, agriculture and forestry in global climate policy analyses"*
- *The referenced (line 570) text in the conclusion has been re-written as: "We believe such hard-linking of CLASH and an IAM would be helpful in examining the optimal role of land-use in mitigating climate change, providing food and biogenic raw-materials for the economy, and in conserving primary ecosystems."*

4.) CLASH represents land use at an aggregate level of ten biomes. However, policy-making happens the level of countries or geopolitical regions and not at the level of biomes. This aspect further limits the usefulness of CLASH for climate policy analysis. This limitation should be reflected and discussed in the manuscript.

- *Yes, this is true. The current version of CLASH is designed for global, single-region IAMs, and by design, such models cannot address questions on the national level. We think that such aggregate-level analyses can provide important insights on the long-term pathways for reaching global climate targets, for example. CLASH could also be easily re-parametrized for different regional divisions, which would be suitable for some specific IAM or policy question.*
- *This was mentioned briefly in the conclusions, but we now rephrased this it more clearly: "CLASH could be easily re-parametrized to alternative geographic resolutions and timesteps to achieve compatibility with a specific IAM; or to represent political borders, which would be necessary for including agricultural, climate, ecosystem protection and other policies relevant for land-use."*

5.) The authors validate the results from CLASH (mostly carbon) against results from LPJ-GUESS. This is fine as such. However, results from CLASH are not compared at all to the results from economic partial equilibrium models such as MAgPIE or GLOBIOM. If CLASH captures certain features of economic land-use models as claimed in the introduction, then the corresponding variables should also be validated against data from these models (e.g. from the IPCC AR6 database) or from FAOSTAT. The same holds true for forest sector models. If CLASH includes features from forest sector models, the corresponding variables should be validated.

- *We now include a comparisons of crop production and wood harvests between CLASH, FAO, SSP scenarios from IAMs (with some of the scenarios using MAgPIE and GLOBIOM) and forest sector models. However, please note that due to the difference in model scopes, this comparison cannot be fully comprehensive. CLASH is a biophysical model and does not*

*therefore include the market mechanisms and policy incentives that the partial equilibrium models have. Yet, by fixing land-use areas in CLASH to a certain SSP scenario (SSP2-4.5 in our case), one can make some comparisons, nevertheless.*

- *These are now presented in three paragraphs and Figure 10 at the end of section 4. CLASH results match FAO statistics generally well and fall mostly to the range of scenarios from other models.*

6.) Do crop yields in CLASH reflect potential yields under optimal management or actual yields under current management? How do yields in CLASH compare to FAO?
To what extent are yield increases due to technological change considered - if at all?

- *The yields in CLASH are based on LPJ-GUESS, which models actual yields following some predetermined management practices, like sowing/harvest dates and nitrogen fertilization (see e.g. www.biogeosciences.net/12/2489/2015/). We did not specify these explicitly in our experiments. We discussed in section 2.7 that LPJ-GUESS seems to be more sensitive to CO2 fertilization than the average of DVGMs, according to the comparison presented by Franke et al. (https://doi.org/10.5194/gmd-13-3995-2020). This is an important driver for the yield changes, along with the warming effect for some biomes, in the SSP2-4.5 scenario we used as the background in this manuscript.*

- *Comparing yields to FAO would not be straightforward, given that CLASH only represents an aggregate crop. However, the comparison of total crop production we produced as a response to the previous question shows that the production volume matches well with FAO statistics, and therefore the CLASH aggregate crop yield corresponds well with the average yield of crops that are being cultivated.*

- *This is now explained more transparently in section 2.7.*

- *The comparison of crop production to FAO and IAM scenarios was added into section 4.*

7.) It seems that existing national climate policies for the land-use sector (e.g. reduced deforestation) are not accounted for in CLASH. Disregarding so-called Nationally Determined Contribution (NDCs) in support of the Paris Agreement further limits the usefulness of CLASH for climate policy analysis.

- *Indeed. As discussed above, such analyses fall outside the scope of CLASH in two respects: it does not consider policies in itself, as it's a biophysical model (although some policies could be introduced externally); and that it does not cover countries separately (although this could be remedied with a new parametrization with a different geographic split between regions).*

- *This is now stated e.g. in the introduction (please see also the responses to the comments above): "What is left out is the product demand, markets and policies that drive land-use decisions in partial equilibrium models […]"*

- *In the conclusions we also now state more explicitly: "CLASH could be easily re-parametrized to alternative geographic resolutions and timesteps to achieve compatibility with a specific IAM; or to represent political borders, which would be necessary for including agricultural, climate, ecosystem protection and other policies relevant for land-use."*

8.) Likewise, it seems that existing protected areas (IUCN WDPA, https://www.protectedplanet.net/en/thematic-areas/wdpa?tab=WDPA) are not accounted for in CLASH but would be needed for meaningful climate policy analysis.

- *As above, such policies are not an inherent part of CLASH, but could be given as external constraints. We have now clarified this in the revised manuscript, as indicated above.*

9.) correct typos: "DVGMs" -> DGVMs

- *Thank you for pointing out these typos. These have been corrected in the revised manuscript.*

**Reviewer #2**

Overall I agree with the authors that this paper presents a modeling system that addresses an open area in integrated assessment modeling. The descriptions are concise and informative, and the demonstrations all look reasonable. I had a number of questions and minor requests for revisions that I'll organize in the order that I encountered them in the manuscript.

- *Thank you. We are glad the Reviewer found the model's focus relevant, descriptions informative and demonstrations reasonable. We are grateful for the questions and suggestions, which helped us to improve the manuscript further. All suggested changes were made to the revised manuscript.*

Line 100: "carbon concentration, as these variables standard outputs of many IAMs" please replace with "carbon dioxide concentration, standard outputs of IAMs"

- *Indeed, thank you for noting this. The text has been updated following this suggestion (although we added 'many' in front of 'IAMs', as not all models provide this output).*

Line 101: "changes in these variables serve as a proxy for the changes on local climatic factors" - I thought that the opposite was true, that realized climatic conditions, precipitation in particular, are regionally heterogeneous and not linearly related to the global average surface temperature and $CO_2$ concentrations. Can this statement be further clarified, or sourced in the literature?

- *Yes, this needs to be stated more clearly. The LPJ-GUESS runs use gridded data of temperature and precipitation that comes from different Earth system models, and thus the LPJ-GUESS runs consider the regional heterogeneity in growing conditions and also how climate change affects the regions differently. Then, as CLASH emulates these LPJ-GUESS simulations, the CLASH parametrization for the different biomes consider this regional heterogeneity (also including intra-biome heterogeneity). Yet, some detail is obviously lost when the gridded results are aggregated into the level of the ten biomes used in CLASH.*

- *We now explain this more clearly: "Climate change affects vegetation growth through changes in local factors, such as temperature and precipitation. In CLASH, vegetation growth and other ecological processes emulate results from LPJ-GUESS, which is run on a spatial grid in different climate scenarios and thus accounts for the regional differences in growing conditions in current and future climates. The biomes in CLASH are large and cover somewhat heterogeneous conditions and responses to climate change (e.g., some parts of a tropical biome may become drier, while others get wetter). The CLASH parametrizations thus depict the average conditions within each biome. To account for climate change, the parametrizations are done as a function of global mean temperature and carbon dioxide concentration, standard outputs of many IAMs. These serve as proxies for the changes on local climatic factors, which are nevertheless modelled explicitly in LPJ-GUESS."*

One of the weaknesses I see in the existing IAM information flow right now is that the $CO_2$ concentrations are estimated in simple climate models (e.g., MAGICC) using an equation that estimates enhanced $CO_2$ uptake by the terrestrial biosphere as a function of ambient $CO_2$ concentrations (i.e., the $CO_2$ fertilization response). The additional uptake in turn reduces the estimated $CO_2$ concentrations. However this additional carbon, which is understood to be taken up into the biomass of vegetation and soils, isn't actually represented in the IAMs' land use modules. It looks like this model explicitly represents that aspect of the dynamics, but also it seems that the $CO_2$ concentrations in CLASH are exogenous, and that the dynamics represented within the model aren't passed to a simple climate model, where they could replace the existing $CO_2$ fertilization response function. Can the authors comment on whether the outputs of CLASH are used to provide an

endogenous and bottom-up estimate of the global terrestrial biosphere's CO2 fertilization response which can then be used to revise the estimated atmospheric CO2 concentrations?

- *Thank you, this comment is right on point and shows that the interaction between CLASH and an IAM needs to be elaborated.*
- *The $CO_2$ concentration is given exogenously to CLASH, as you wrote. The vegetation growth in CLASH then reacts to this, so that one can derive the net exchange of carbon between the atmosphere and biosphere, also accounting for the fertilization effect.*
- *Some IAMs have a simple, built-in climate module that can calculate $CO_2$ concentration and pass this to CLASH (c.f. the DICE climate module and similar approaches). If CLASH is linked to such IAM, CLASH can represent the net exchange of carbon between the atmosphere and the terrestrial biosphere. But as you correctly point out, the IAM's climate module should not then consider this exchange (but only the exchange between the atmosphere and oceans), as CLASH accounts for it already. Although this point concerns more the climate modules of IAMs, it should be mentioned in the manuscript to avoid possible errors if some other researchers decide to link CLASH with their IAMs.*
- *We now note this explicitly in the conclusions: "If the IAM has a built-in climate module, it can provide the future $CO_2$ concentration and temperature change for the CLASH ecological module. However, in such setup, the climate module should not consider the carbon exchange between atmosphere and terrestrial ecosystems, as CLASH accounts for their carbon stocks and the fertilization effect from elevated $CO_2$ concentrations. The change in carbon stocks can then be used to calculate the net exchange of $CO_2$ with the atmosphere, from both managed and primary terrestrial ecosystems."*

Section 2.7 - in general, when referring to masses and densities, please distinguish between carbon, total vegetation, and dry vegetation. I generally found the terminology to be a bit vague. For instance, in Table 1's caption and conversion factor unit, I'd recomment using "kgC/m2" instead of the currently written "kg/m2".

- *Thank you for pointing this out. One can often be blind to such ambiguities in one's own text. We have now clarified the units (e.g. separating between kgC and kgDM) in section 2.7 and Table 1. We also removed the explanatory footnote and included the breakdown of the conversion factor in the main text, which should make the text clearer to read.*

Section 2.8 - this is much more detail on the livestock side than I'd have expected for a land/carbon model, particularly given that process-based IAMs already have pretty detailed representations of

the inputs and outputs of livestock production. It is noted in the text that the carbon in animal biomass is trivial and as such is omitted. So then, can the authors clarify in this section the purpose of the livestock module, and how it would interface with the livestock representation in an IAM in order to ensure consistency? It just seems strange right now that livestock production, which is economic in nature, would be represented in this model which doesn't represent economic considerations.

- *The main reason is that livestock uses vast amounts of land area, both directly and indirectly through feed production, and also produces notable amounts of $CH_4$ and $N_2O$ emissions. Although the scope of CLASH is biophysical, thereby excluding the economic and policy aspects of land-use, we wanted to represent the land-use and GHG emission impacts of livestock. Otherwise, the model would not be able to represent why such large areas are allocated for pastures. As an example for the relevance of this, the demonstration of the model provided in section 5 analyzed how land-use, carbon stocks and agricultural non-$CO_2$ emissions react to different demand scenarios of livestock products. Although this is a purely biophysical consideration, it already can provide insight into the role of livestock for climate.*
- *We now provide this motivation in the beginning of section 2.8: "CLASH includes a representation of livestock to account for the land area needed for cattle grazing and cultivating livestock feed crops, and also for the $CH_4$ and $N_2O$ emissions from livestock management."*
- *Should CLASH be integrated into an IAM that already has a bottom-up representation of livestock, one has the options of disabling the livestock representation from CLASH, the IAM, or otherwise connecting the two.*

278 - please move the disambiguation of PFT up from line 278 to 277.

- *Thank you for noting this. We introduced PFTs already on line 210, so the abbreviation will be used from thereon.*

Figure 7 - do the crop yields take management practices into consideration? These are the main drivers of historical and future projected yield change, and I'd think they should be an exogenous input from an IAM or other model. The yield changes are so large it looks like they might be considered, but the text suggests otherwise.

- *LPJ-GUESS includes some management practices, like sowing/harvest dates and nitrogen fertilization (see e.g. www.biogeosciences.net/12/2489/2015/). We did not specify these*

*explicitly in our experiments. Also, CLASH does not enable the optimization of agricultural practices to improve yields. The model seems to be more sensitive to CO2 fertilization than the average of DVGMs, according to the comparison presented by Franke et al. ([https://doi.org/10.5194/gmd-13-3995-2020](https://doi.org/10.5194/gmd-13-3995-2020)), and this is the main driver behind the yield changes, along with the warming effect for some biomes, in the SSP2-4.5 scenarios that we used as the background in this manuscript.*

- *These are mentioned more explicitly now in section 2.7.*

In the end I'm unclear about what is envisioned for linking with IAMs. This is claimed to be the main value added in the introduction, but then all of the details are left vague in the subsequent sections. Here are some of the questions I have, which don't all need to be answered directly and completely in a revised manuscript, but some information would be helpful.

Which IAM(s) is (are) CLASH intended to be "hard linked" to?

Which variables would be exogenous, imported from the IAM?

What would be the information flow from CLASH to the IAM?

Would it be used to revise the CO2 concentrations of the simple climate model?

- *Indeed, this might have been bit vague, partly because the question over IAMs is not inherently a feature of CLASH itself. We have integrated this with the SuCCESs IAM (a new bottom-up IAM we have recently developed) and one of the authors is carrying out the linkage with ICICLE, a simpler top-down IAM. Neither of the works have yet been fully published, however. Also, the potential uses of CLASH are not limited to these models. Due to this, we saw that mentioning any IAMs explicitly is not relevant for the scope of this paper.*

- *We now discuss the CLASH-IAM linkage in the concluding section: "When integrated with an IAM, CLASH calculates the annual production of land-use commodities (food and other biomass for energy and materials), the change in terrestrial $CO_2$ stocks and GHG emissions from agriculture. The IAM should provide the demand for these products, and any costs, policies and other societal constraints that affect land-use. As such, CLASH would depict land-use from a biophysical perspective, whereas the IAM provides the motivation for how the land should be used and managed."*

---

## Author Response (AR2)

**Response to Reviewers – GMD-2023-146 (revision 1)**

We are happy that the Editor and both Reviewers were satisfied with the revised version of the manuscript. The remaining technical corrections have been carried out. The authors would like to thank both Reviewers and the Editor for their insightful comments during the review process.

**Reviewer #1**

The authors did a great job in a) answering all my previous comments in detail and b) revising the manuscript accordingly.

- *Excellent! Thank you again for the good and constructive feedback for improving the paper.*

Pending the following technical correction, I recommend publication of the paper:

Figure 10b: please correct color for FSM scenarios

- *Thank you for noting the color difference between the figure and the legend. This has now been corrected.*

**Reviewer #2**

Can the authors double check the forestry production data from FAO in Figure 10? The quantities seem about 50% of what I see reported in FAOSTAT.

- *Apologies, the figure caption's descriptor for the data was slightly misleading. (This was stated in the correct way in the main text.) The presented FAO statistics and model results represent industrial roundwood harvests, although the text stated 'Roundwood harvests'. The difference is that the former does not include wood fuel, but only sawlogs, pulpwood and other industrial roundwood. This has now been corrected to the figure caption and the y-axis.*

Also the right panel includes a gray series that isn't listed in the legend.

- *Thank you. Reviewer #1 also noted the color difference between the figure and the legend. This has now been corrected.*

Also, in lines 596-599, I wasn't suggesting (in round 1 review) that an IAM paired with CLASH should turn off its simple climate model's CO2 fertilization response. I was just asking if this model was designed to replace that function in a pairing. Because it is not (at least at this stage of development), the IAMs should not turn off the CO2 fertilization function, unless a team is interested to devise a method whereby the terrestrial CO2 flux would be estimated from the CLASH outputs. For revised text, I'd suggest:

"However, in such a pairing, note that there would likely be a discrepancy between the CO2 fertilization function in the IAM's simple climate model, and the CO2 fluxes represented structurally by CLASH. Future researchers may seek to replace the CO2 fertilization function in the simple climate model with the relevant outputs from CLASH, in order to improve the simple climate model's estimates of CO2 concentrations."

- *Thank you for clarifying the comment in the first round of reviews. We modified the suggested text a bit further, arriving at (including also here the paragraph's first sentence): "If the IAM has a built-in climate module, it can provide the future CO2 concentration and temperature change for the CLASH ecological module, while CLASH can calculate the net carbon exchange of terrestrial ecosystems. However, in such a setup, it is important to note a likely discrepancy between the climate module's carbon cycle and the carbon stocks represented by CLASH. Particularly, the climate module should not represent the carbon exchange between atmosphere and terrestrial ecosystems, as CLASH accounts for terrestrial carbon stocks and the fertilization effect from elevated CO2 concentrations. Further model development might be needed to replace relevant parts of the IAM climate module (or an external simple climate model) with associated outputs from CLASH to ensure consistency between the two parts."*